# Antibiotic Resistance in Plant Pathogenic Bacteria: Recent Data and Environmental Impact of Unchecked Use and the Potential of Biocontrol Agents as an Eco-Friendly Alternative

**DOI:** 10.3390/plants13081135

**Published:** 2024-04-18

**Authors:** Tarequl Islam, Md Azizul Haque, Hasi Rani Barai, Arif Istiaq, Jong-Joo Kim

**Affiliations:** 1Department of Microbiology, Noakhali Science and Technology University, Sonapur, Noakhali 3814, Bangladesh; tarequembg@gmail.com; 2Department of Biotechnology, Yeungnam University, Gyeongsan 38541, Gyeongbuk, Republic of Korea; azizul@ynu.ac.kr; 3School of Mechanical and IT Engineering, Yeungnam University, Gyeongsan 38541, Gyeongbuk, Republic of Korea; hrbarai@ynu.ac.kr; 4Department of Pediatrics, Division of Genetics and Genomic Medicine, Washington University School of Medicine, St Louis, MO 63110-1010, USA

**Keywords:** plant pathogenic bacteria, biocontrol agents, agriculture, environmental impact, bacterial endophytes, antibiotic-resistant genes (ARGs)

## Abstract

The economic impact of phytopathogenic bacteria on agriculture is staggering, costing billions of US dollars globally. *Pseudomonas syringae* is the top most phytopathogenic bacteria, having more than 60 pathovars, which cause bacteria speck in tomatoes, halo blight in beans, and so on. Although antibiotics or a combination of antibiotics are used to manage infectious diseases in plants, they are employed far less in agriculture compared to human and animal populations. Moreover, the majority of antibiotics used in plants are immediately washed away, leading to environmental damage to ecosystems and food chains. Due to the serious risk of antibiotic resistance (AR) and the potential for environmental contamination with antibiotic residues and resistance genes, the use of unchecked antibiotics against phytopathogenic bacteria is not advisable. Despite the significant concern regarding AR in the world today, there are inadequate and outdated data on the AR of phytopathogenic bacteria. This review presents recent AR data on plant pathogenic bacteria (PPB), along with their environmental impact. In light of these findings, we suggest the use of biocontrol agents as a sustainable, eco-friendly, and effective alternative to controlling phytopathogenic bacteria.

## 1. Introduction

Phytopathogenic bacteria are responsible for causing plant diseases and can have adverse effects on a wide variety of crops, resulting in economic losses and negative environmental impacts. Research on phytopathogenic bacteria aims to deepen our understanding of their taxonomy, genetics, and plant pathology, including the mechanisms underlying plant diseases. Recent advances in genomics and molecular plant pathology, along with the emergence of new bacterial plant diseases, have led to rapid evolution and change in this field [1,2].

The use of antibiotics to treat bacterial plant diseases is limited, and they are typically reserved for high-value fruit crops due to concerns regarding antimicrobial resistance (AR). Pathosystems where antibiotics have been used for an extended period have seen the emergence of multi-drug-resistant (MDR) infections [3], which are more severe and have a significant financial impact [4,5]. In fact, AR is projected to cost the global economy up to USD 100 trillion in lost productivity and result in 300 million premature deaths by 2050 [6].

Understanding how bacteria survive is critical to developing novel methods for combating plant diseases. One such strategy is the development of persistent cells, which are a tiny subset of phenotypic variants displaying multi-drug tolerance without undergoing genetic change. They can deactivate metabolic processes, which are interfered with by antimicrobials, allowing them to survive treatment. However, they may also be responsible for disease recurrence [7].

Biological control is a promising method for inhibiting plant pathogens, enhancing plant immunity, and altering the environment through the effects of advantageous microbes, substances, or healthy cropping practices [8,9,10,11,12]. Epiphytic bacteria, fungi, and bacteriophages have all been successfully employed to control various plant diseases and offer possible alternatives to antibiotics [13,14,15,16,17,18]. Additionally, expanding the genetic resources accessible to breeders through genetic alteration and genome editing is an efficient and sustainable method of managing plant diseases [19].

This review discusses the top five phytopathogenic bacterial genera, the antibiotics used to control them, and their recent AR profile with variations in resistance over time. We also address the mechanism of AR and its spread to food-borne pathogens, as well as the impacts of AR on plants, humans, and the environment. Finally, we discuss the use of bacterial, fungal, and viral biocontrol agents to control phytopathogenic bacteria.

## 2. Plant Bacterial Pathogens and Their Control by Antibiotics

### 2.1. Plant Pathogens and Diseases

There are many bacterial pathogens residing in plants. They have their own biochemical properties and modes of invasion. All of these bacterial pathogens cause worldwide losses of over USD 1 billion every year [7,20]. It is in fact difficult to categorize plant bacterial pathogens based on their pathogenicity. However, the journal *Molecular Plant Pathology* conducted a survey—enlisting the participation of bacterial pathologists—in order to categorize plant pathogens and rank them based on their severity [20]. They ranked the top ten bacterial species based on that survey [20]. However, in this review, we briefly discuss the top five genera of plant bacterial pathogens based on the abovementioned survey. Hence, the plant bacterial pathogens subject to discussion in this review include (a) *Pseudomonas*; (b) *Ralstonia*; (c) *Agrobacterium*; (d) *Xanthomonas*; and (e) *Pectobacterium*.

#### 2.1.1. *Pseudomonas* spp.

One of the most prevalent plant diseases infecting the phyllosphere is *P. syringae*, but its taxonomy is argumentative [21]. Currently, twenty species are available in the taxonomy, and some more new species have been proposed (e.g., *P. viridiflava*) [21,22]. *P. syringae* can exist as an epiphyte on the plant’s surface [20]. It is a rod-shaped, Gram-negative bacterium with polar flagella [23] and the top most studied plant pathogen, which is ranked as number one plant pathogenic bacterium by plant pathologists [20]. *P. syringae* pathovars cause different types of diseases, including bacteria speck in tomato, halo blight in beans, bleeding cancer in different woody plants, and bacterial cancer in kiwifruits. It can colonize in different plant tissues, including seeds, leaves, fruits, and bark. Infections with *P. syringae* include symptoms such as chlorosis, cancer, blight, and water-soaked lesion [24]. The economic losses caused by *Pseudomonas* in agriculture can vary widely depending on factors such as the type of crop affected, the severity of the infection, and the effectiveness of the disease management system [25]. *P. syringae* pv. *actinidiae* causes bacterial canker in kiwifruits and leads to enormous economic losses in Italy, New Zealand, France, Spain, Portugal, Chile, South Korea, and Japan. However, it is difficult to precisely estimate the total economic losses caused by *Pseudomonas*.

The 13 phylogroups of *P. syringae* strains were identified between early branching and canonical lineages. The canonical lineages comprise multiple plant-specialist phylogroups with conserved virulence-associated and phenotypic characteristics [26]. *P. syringae* has been separated into over 60 pathovars based on host isolation, host range, and other characteristics [27]. External environmental factors have a significant impact on *P. syringae* infection [24]. *P. syringae* encounters the apoplast—a potentially carbohydrate-rich but fiercely protected dwelling area for bacteria—after entering the plant [24].

*P. syringae* uses different types of virulent factors to attack plants. A variety of host–pathogen interactions are found to be associated with *P. syringae* infection depending on its virulence properties. These interactions include toxins, ice nucleation proteins, secreted effectors, and antimicrobial resistance proteins. Type III secretion system is the best studied virulence-linked factor, which is involved both in restricting and promoting specific host–pathogen interactions [28,29,30,31]. *P. syringae* uses four primary toxins: coronatine, phaseolotoxin, syringomycin, and tabtoxin [31]. The mode of action of these toxins depends on the nature of the toxins [31]. The ice nucleation proteins of *P. syringae* cause frost damage to crops [23].

By detecting pathogen-associated molecular patterns (PAMPs), plants have developed a defense mechanism (stomatal closure) to prevent bacterial ingress through stomata. A recent study identified the resistant variant in wild and ornamental cherry toward *P. syringae* [32]. Understanding the genetic mechanism of resistant variants would be a promising step toward understanding the resistance mechanism against this pathogen.

#### 2.1.2. *Ralstonia* spp.

Soil-residing *Ralstonia* are one of the world’s most dangerous phytopathogenic bacterial species, causing bacterial wilt disease in over 400 plant species and posing a serious threat to agriculture [20,33]. Many commercially significant crops, including tobacco, tomato, and potato, are infected by this bacterium [34]. Tomato bacterial wilt caused by *Ralstonia* has the potential to wipe out the entire crop [35]. *Ralstonia solanacearum* is responsible for yield losses of approximately 80% and 90% in eggplant and tomato, respectively [36]. Herbaceous plants comprise the majority of hosts. Yield losses of potato caused by *R. solanacearum* may vary from 33% to 90% [37]. In India, R. solanacearum causes 2–95% crop damage depending on the season and cultivars [38]. The pathosystem in the case of woody hosts is substantially less well understood [39].

*R. solanacearum* strains are classified as a heterogeneous group of species with four phylotypes, five races, and six biovars based on their geographic origin [34]. Recently, a new proposal has been suggested by a taxonomic and nomenclatural update, namely that the *R. solanacearum* species complex (RSSC) should include three distinct species: *R. pseudosolanacearum* (formerly phylotypes I and III), *R. solanacearum* (IIA and IIB), and *R. syzygii* (formerly phylotype IV and blood disease bacterium) [40,41].

Soil- and water-borne *Ralstonia species* enter the host through the roots, causing wilting by colonizing the xylem vessels in large numbers, resulting in vascular malfunction [33,42]. The extremely high amount of virulence and pathogenicity factors they manufacture has been linked to the damage they inflict [43,44,45]. The speed and severity of wilt symptom development is determined by the host’s age, health, and nutritional state, as well as environmental factors and pathogen aggressiveness [39]. Both biotic and abiotic factors are responsible for the infections caused by *Ralstonia* species. Abiotic stress factors—such as inadequate root systems, improper planting procedures, bad site conditions, or infections with primary pathogens, such as *Ganoderma philippii*—appear to be required for infection to take place [39].

Physical treatments are ineffective; crop rotations are often impractical; and the pathogen displays high aggressiveness and endurance in adverse environmental conditions. No management method appears to be fully advisable against bacterial wilt because crop protection chemicals do not provide sufficient control and usually have a negative impact on the environment and/or human health, favor the emergence of resistance, and are expensive [46,47,48].

#### 2.1.3. *Agrobacterium* spp.

Over a century ago, *Agrobacterium* was recognized as the cause of crown gall—a plant tumor [49]. *Agrobacterium* bacteria are soil-borne bacteria, which can reside in the rhizosphere of plants, and they can be harmful if they carry the tumor-inducing (Ti) plasmid. T-DNA genes are expressed by the infected plant, resulting in hormone synthesis and crown gall disease (uncontrolled proliferation of plant cells) [50]. *Agrobacterium* is one of the few bacterial phytopathogens, which uses its host to build a niche rather than killing it [51]. These bacteria are polyphyletic pathogens with vast and narrow host ranges [52]. Crown gall causes approximately 5% of global crop losses in over 2000 susceptible plants [53]. It affects around 100 greenhouse and nursery species, causing USD 16.2 billion in annual loss in the United States [54]. Their taxonomic and phylogenetic classifications have yet to be determined. *A. tumefaciens* can survive in soil and in the presence of plants. Many attributes undergo major modifications in expression during the move from soil to a host environment. Features associated with a motile, individual lifestyle are downregulated during growth in planta or in the rhizosphere—which is imitated by cultivating *A. tumefaciens* under acidic conditions (pH = 5.5)—while those related to sessile, communal activity are upregulated [55].

Opines are amino-acid/sugar or organic-acid conjugates, which are specially utilized as nutrition by agrobacteria with the Ti plasmid. Around 40 different varieties of opines have been identified; some of them cause the Ti plasmid to be transferred from one bacterium to another, increasing pathogenicity and contributing to pathogenic bacteria’s persistence in the environment [56,57]. Despite the fact that plants launch defense mechanisms against *Agrobacterium*—which secrete chemical molecules, such as salicylic acid (SA), jasmonic acid (JA), or ethylene—the bacterium has been found to escape such barriers and establish long-term residence in tumors [58,59]. Auxin and cytokinin syntheses are induced via T-DNA incorporation into the plant genome. Cell proliferation and tumor growth are accelerated by high doses of these two phytohormones. Ethylene has two key functions in the tumor: it lowers the diameter of plant vessels around the tumor to keep it hydrated, and it induces the production of abscisic acid [60].

These pathogens must circumnavigate a wide range of environmental heterogeneity. The microbes’ success within the host is shaped by host responses, and each environment provides different nutrient availability and microbiota. Transferring between these habitats usually necessitates the pathogens changing their physiology and behavior, as well as a shift in the population’s evolutionary selective forces, opening the door to ecological dilemmas in which features that are advantageous in one habitat may be deleterious in another [61,62].

#### 2.1.4. *Xanthomonas* spp.

*Xanthomonas* spp.—ranking at number four in the top ten plant pathogens—is a Gram-negative, yellow-pigmented bacterium, which comprises 35 species [63] and invades around 400 plants, including rice, citrus, banana, cabbage, tomato, pepper, bean, and so on [64]. It is generally a rod-shaped obligate aerobe with a single polar flagellum, and its optimal growth temperature range is 25 °C–30 °C [65]. The species of *Xanthomonas* show high specificity for the host and tissue—invading the xylem and the intercellular spaces of the mesophyll parenchyma tissue—and can be distinguished into pathovars [66]. Initially, the bacteria grow on the leaf surface (epiphytic growth) and enter the vascular system or mesophyll parenchyma tissue through wounds or other natural openings [1,4]. On the leaf surface, the micro-organisms produce a complex community. There are very few studies focusing on this aspect. However, in order to survive in this complex community structure, *Xanthomonas* has to utilize its own multifarious mechanism. Like many other bacteria, *Xanthomonas* produces its own single niche on the leaf surface [67]. It produces type IV and VI secretion to overcome the difficulties produced by other communities. *Xanthomonas* produces antimicrobial peptides through type IV secretion, which kills the Gram-negative bacteria [68]. *X. citri* overcomes the challenge imposed by the predator *Dictyostelium* via type VI secretion [69].

The most common diseases caused by *Xanthomonas* include bacterial blight of rice, bean, and cassava, citrus cancer of citrus plants, bacterial rot of crucifers, gumming disease and leaf scald of sugarcane, enset wilt of banana, and so on [65,66,70,71]. These diseases result in reduced crop yield, lower quality produce, increased production cost due to disease management, and—in severe cases—complete crop loss [65,66,70,71]. Overall, the economic losses from banana *Xanthomonas* wilt were estimated at USD 2–8 billion over a decade in east and central Africa [72,73]. These economic losses depend on the regions, plants, and types of diseases. Citrus canker caused by *Xanthomonas* causes annual losses of over USD 1 billion [74]. The environment, contaminated seeds, weeds, and contaminated plant debris are the main sources of *Xanthomonas* transmission [65]. Relatively higher temperature (25–35 °C) and humidity facilitate *Xanthomonas* in the invasion of host plants [75,76,77]. Plant stomata and hydathodes are opened at higher humidity in order to withstand water [78]. The opening of hydathodes at night and at high humidity is thought to be a mechanism of *Xanthomonas* infection and dispersal [79]. Heavy wind and storm are regarded as the main sources of *Xanthomonas* dispersal. Laboratory examination of *X. citri* and *X. alfalfae* demonstrated that there is a strong correlation between heavy wind and pathogen dispersal to the nearby host plant [80]. Windbreak application was successful in reducing the outbreak caused by the pathogen [81]. Strong surveillance after heavy storm helped detect the possible outbreaks [81]. People and farming tools are two major means of dispersal of the pathogen. Banana and plantain pathogen *X.campestris* pv. *musacearum* was reported to spread via contaminated agricultural tools, whereby the pathogen could survive 2–3 weeks on the tools [82]. Many *Xanthomonas* dispersions were reported to be associated with infected seeds [83,84]. The pathogen can reside inside the seeds for several weeks [83]. Research on *X. campestris* and *X. oryzae* revealed that their duration of stay varied depending on the hosts [84]. A laboratory experiment on *Xanthomonas* showed that infections with the pathogen were apparent in plants when inoculated at different development stages, but it could only colonize into the seeds when inoculated during the flowering season [85]. A further study of *X. campestris* by the group revealed that it could successfully colonize into the outer layer of seed coat and the endosperm and embryo [86].

Functional and comparative genomics revealed extensive genome variations within the *Xanthomonas* genus, which helped it fit with diverse plant hosts and tissues. These large-scale variations are mainly due to the presence of external plasmids and insertion sequences (ISs) within the population [64]. Some *Xanthomonas* species consist of genomes similar to alphaproteobacterial, betaproteobacterial, and gammaproteobacterial origin, and some others are related to the Archaea, Eukarya, and viruses. The presence of genes similar to other bacterial and phenotypically distinct organisms may be due to horizontal gene transfer [87,88]. *Xamthomonas* contains a single circular chromosome, ranging from 4.8 Mb to 5.3 Mb, with 60% GC content. *X. fastidiosa* has a genome around 2.7 Mb, which only infects the host plant xylem and is transmitted by the insect vector. *X. campestris* pv. *campestris* and *X. oryzae* pv. *oryzae* have an extended genome size, which can colonize in seeds and survive in dead plants in soil [89]. Many of the genes in *Xanthomonas* associated with virulence originate from external plasmids, ranging from 2 Kb to 183 Kb. The sequencing of some of these plasmids revealed the presence of genes responsible for type III and type IV secretions (T4SS). The pXCV183 plasmid of *X. euvesicatoria* encodes Dot/Icm T4SS, which is similar to the human pathogen *Legionella pneumophila* [90].

#### 2.1.5. *Pectobacterium* spp.

*Pectobacterium* is a genus of Gram-negative, facultative anaerobic, non-spore-forming, and extracellular-pectinase-producing plant pathogenic enterobacteria belonging to the Proteobacteriaceae family [91,92]. Previously, it was known as *Erwinia*—for instance, *Erwinia carotovora* ssp. *carotovora*, which is now known as *Pectobacterium carotovora ssp. carotovora* [91]. Members of the *Erwinia* genus were divided into three genera: *Erwinia*, *Pectobacterium*, and *Brenneria* [93]. *Pectobacterium* invades different parts of plants, causing blackleg, soft rot, and aerial stem rot in carrot, tomato, cabbage, potato, and many other plants all over the world [91,92,94]. The pathogenesis of *Pectobacterium* mainly depends on the production and secretion of enormous amounts of extracellular enzymes, such as pectate lyases, proteases, polygalacturonases, and cellulases, which cause cell wall degradation, tissue softening, and rotting, resulting in plant death [92,95].

Some abiotic factors—such as temperature, free water, and available oxygen—play important roles in disease progression [95]. *Pectobacterium* shows varying pathogenicity with the change in temperature. Maximum pathogenicity was recorded between 28 °C and 30 °C. Decreased pathogenicity was found between 24 °C and 37 °C [96]. Proteases, polygalacturonases, cellulases, and pectolytic activity decreased with the increase in temperature, whereas pectate lyases did not show a significant change with the increase in temperature [96]. However, the disease symptoms caused by *Pectobacterium* vary depending on the hosts and bacterial species [95]. *P. carotovorum ssp. carotovorum* has wide-ranging hosts, including carrots, potatoes, lettuce, cabbage, onions, and so on, all over the world [97,98,99]. *Pectobacterium atrosepticum* and *P. carotovorum ssp. carotovorum* are the main causes of soft rot and/or blackleg in temperate areas [99,100,101].

The *Pectobacterium* genus is composed of heterogeneous strains. *Pectobacterium* species are also divided into different subspecies. The *P. carotovorum* species has four subspecies—*carotovorum*, *brasiliense*, *odoriferum*, and *actinidiae*—which have differing appearances [102]. A total of 265 *Pectobacterium* strains were reported from 1944 to 2020 [103]. Thirteen species of *Pectobacterium* have been identified to date [96,102,103]. *Pectobacteruim* shows substantial genome variations among the species, although most virulent genes remain conserved. A pangenome analysis of *Pectobacterium* shows a dynamic evaluation process via gene loss or gain and rearrangement [104]. These genetic variations are the main causes of the differences in pathogenicity among the strains of *Pectobacterium* [104].

*Pectobacterium* causes enormous economic losses worldwide. *P. parmentieri*—a recently identified bacterium of the Pectobacteriaceae family—is highly pathogenic for the economically important crops and the causative agent of soft rot of potato. It can grow and infect in a variety of environmental conditions worldwide, including Africa, Europe, North America, and New Zealand. Several genes of this bacterium encode different virulent extracellular enzymes, which cause cell damage in plants [105]. *P. atrosepticum* and *P. carotovorum ssp. carotovorum* are the main causes of soft rot [100,101]. *P. atrosepticum* was recorded as a substantial threat to potato production in Northern Ireland [106]. *Pectobacterium* and *Dickeya* are responsible for blackleg and soft rot diseases in EU potato production, causing losses of approximately EUR 46 million annually [107]. The losses recorded were 32%, 43%, and 25% for the seed potato sector, the table potato sector, and the processing potato sector, respectively [107].

### 2.2. Antibiotics in Plant Pathogen Control

Plants are known to harbor numerous bacterial pathogens, which exhibit variations in their biochemical properties and modes of invasion. The classification of plant bacterial pathogens based on their pathogenicity is challenging due to these variations. In an attempt to categorize and rank plant pathogens based on their severity, a survey was conducted by the journal *Molecular Plant Pathology* with the participation of bacterial pathologists who ranked the top ten bacterial species [20].

Bacterial plant diseases are difficult to manage due to the large populations of bacterial pathogens in sensitive plant hosts and the limited availability of bactericides. Antibiotics have been widely used as an alternative to control bacterial plant diseases since the 1950s, as they are effective in reducing bacterial population size and preventing disease outbreaks in the absence of long-lasting and powerful host disease resistance. Oxytetracycline and streptomycin are currently the most frequently used antibiotics in plants [108]. In addition, kasugamycin, gentamicin, and oxolinic acid are also used to control bacterial plant pathogens [108]. The amount of antibiotics used in plant agriculture is minimal compared to other applications. Plant-grade antibiotics are usually created as powders with 17%–20% active ingredients, which are then dissolved or suspended in water at concentrations between 50 and 300 ppm before application [108,109].

However, the use of antibiotics in agriculture has increased in recent years, and there are now more specific uses of antibiotics against pathogens [109]. Eleven antibiotics are recommended for use in crops (often in combination), with significant regional variation in their application [109]. Antibiotics are prescribed for a variety of issues, and they are frequently applied as a prophylactic spray to prevent or control low levels of bacterial illness. Table 1 shows the most frequently used antibiotics against specific PPB. The European Union has prohibited the use of antibiotics in plant disease control due to concerns regarding their potential effects on human health, although antibiotics are still used in agriculture [3].

### 2.3. Antibiotic Resistance Profile of Plant Pathogens

The global health problem of antibiotic resistance (AR) is exacerbated by the movement of micro-organisms and genes between people, animals, instruments, and the environment [111,112,113,114]. Despite numerous barriers prohibiting both bacteria and genes from moving freely, pathogens frequently acquire new resistance components from other species, making it more difficult to prevent and treat bacterial illnesses. The creation of new disease resistance traits through rare and difficult-to-predict evolutionary processes could have significant effects [115].

Although antibiotics’ use in agriculture is lower than in human and animal healthcare, their increased use has contributed to the creation of new resistant strains of bacterial pathogens. For example, *C. michiganensis* showed both spontaneous and induced resistance to streptomycin due to multi-drug efflux pumps or enzymes—which inactivate streptomycin—and *rps*L mutation [116]. Another study found that higher use of streptomycin sprays in citrus orchards resulted in higher resistance of the *Xanthomonas smithii* subsp. *citri* [117]. Similar resistance was observed with other antibiotics, where bacteria acquired resistance due to chromosomal mutation or resistance gene acquisition.

In this review, we attempted to summarize the AR data from previous research on PPB. Table 2 shows the AR pattern of PPB worldwide over 17 years, while Table 3 shows the higher minimum inhibitory concentration (MIC) values of antibiotics found against PPB over time. Both of these tables show enormous variation in PPB susceptibility to antibiotics. The locations of places where the studies on antibiotic resistance discussed in this review were carried out are shown in Figure 1A. The study areas are marked red in the figure, which includes thirteen countries. This VOSviewer 1.6.20 network analysis combines insights from 16 related research publications on antibiotic resistance of PPB, which were studied in this review, highlighting the complex interplay between antibiotic resistance and bacterial diseases in agricultural settings (Figure 1B). The network underscores the geographical diversity of the research, with studies spanning different regions of the world, and emphasizes the importance of understanding the resistance mechanisms in order to develop effective disease management strategies. The analysis serves as a testament to the collaborative efforts in addressing the global challenge of antibiotic resistance in plant pathogens.

## 3. Mechanisms of Antibiotic Resistance

Organisms have a natural ability to adapt to changing or adverse conditions, including antibiotic exposure, leading to AR [111,112,131]. However, different micro-organisms employ distinct mechanisms to become resistant to antibiotics, and various factors contribute to their resistance [115]. Bacterial pathogens are known for their genetic plasticity, which can cause mutational modifications, transfer of genetic material, or changes in gene expression, resulting in resistance to nearly all antibiotics used in clinical practice [4,132]. Acquisition of resistance plasmids or chromosomal mutations are the two primary mechanisms involved in AR [113,114]. Plasmids bearing resistance genes can result in antibiotic degradation through enzyme synthesis or enzymatic modification, while mutations leading to antimicrobial resistance can alter antibiotic activity by modifying the antibiotic target site, activating harmful molecule excretion, reducing drug uptake, or changing important metabolic pathways [4]. Figure 2 summarizes the mechanisms of AR. Mutation resistance involves the development of mutations in a subset of bacterial cells from susceptible populations, leading to conserved cell viability in the presence of the antibacterial molecule. The resistant bacteria replace the susceptible population. However, mutagenic changes leading to drug resistance may be challenging to maintain for the resistant population and are only sustained in the presence of antibiotics [4].

Co-resistance is another type of resistance mechanism, where resistance to one antibiotic can cause resistance to another or to a heavy metal [133]. A recent study investigated gentamicin and arsenite co-resistance and the putative molecular mechanisms [128]. Thirty-two gentamicin-resistant (GR) isolates out of thirty-three were resistant to arsenite and carried integrative and conjugative elements (ICEs) and ars-operon-related genes, indicating that arsenite resistance might have developed in GR lineages [128].

### 3.1. Mechanism of Spread of Antibiotic Resistance to Food-Borne Pathogens

Antibiotic resistance is a major public health threat, and its spread to food-borne pathogens is a serious concern. Antibiotic use creates an environment in which micro-organisms must adapt, leading to the emergence of resistant bacteria. While antibiotics can eliminate some disease-causing bacteria, they also kill beneficial bacteria, which protect us from infection. Resistant bacteria proliferate and can transfer their resistance mechanisms to other pathogens via plasmids carrying antibiotic-resistant genes (ARGs), including phytopathogens, soil bacteria, and zoonotic bacteria, which are sometimes present in the surrounding environment and in the food chain [134].

Antimicrobial use is essential for protecting human health, but it also poses a risk of developing antibiotic-resistant micro-organisms [131]. Bacteria can adapt to antibiotics and develop resistance, and the indiscriminate use of antibiotics in agriculture is contributing to the emergence of new strains of resistant bacteria, including plant pathogens, which pose a significant threat to agriculture. In addition, the antibiotics used in agriculture can contaminate the water system, leading to the development of antibiotic-resistant micro-organisms in aquatic microbiota, which can eventually be transferred to humans and animals upon consumption. As the plant microbiomes—particularly rhizosphere micro-organisms—are intimately linked to soil, water, and the atmosphere, ARGs can proliferate across the ecosystems and become dangerous when they reach human bodies through bacterial infection [2]. Using ATP binding cassette transporters, the root exudates generated by root cells may be sent to the plant rhizosphere where they can attract particular microbial populations. ARGs can be produced by a recombination of genes or mutations caused by the antibiotics found in root exudates [2]. Tn5393—a well-known example of an ARG vector in PPB—was detected outside of PPB in *Salmonella enterica* and *Klebsiella pneumoniae*, and its strikingly similar variants (carrying streptomycin resistance genes) developed into complicated connections with other MGEs and ARGs [1].

The use of antibiotics in fish farms can also contribute to the spread of antibiotic resistance [135,136,137,138,139]. Insects, birds, and animals that feed on agricultural plants and seeds can further disseminate resistant bacteria and genes. Furthermore, migratory birds have been shown to spread multiple antibiotic-resistant pathogens in various studies, highlighting the widespread nature of this problem [140,141,142]. Antibiotic-resistant pathogens have also been reported in chickens, further exacerbating the issue [113].

As a result, food-borne bacteria and pathogens are acquiring resistance genes, leading to the emergence of untreatable diseases. AR genes from different edible parts of plants have been reported, as summarized in Table 4, and Figure 3 illustrates the mechanisms of spread of antibiotic resistance throughout the food chain, starting from the agricultural field. Moreover, a recent study reported the presence of antibiotic resistance in plant-growth-promoting bacteria [143]. Therefore, urgent action is needed to address this growing problem and ensure the continued effectiveness of antimicrobial therapies [113,135,136,137,138,139,140,141,142].

### 3.2. Impacts of Antibiotic Resistance

Antibiotics are widely used as growth promoters in animals and for treating infections in both humans and animals, including aquaculture [156,157,158]. Some antibiotics used in agriculture and aquaculture—such as erythromycin, gentamycin, enrofloxacin, neomycin, and streptomycin—are structurally related to those used to treat human infections [159]. As a result, antibiotic-resistant bacteria in agriculture and aquaculture can enter the human food chain, causing cross-resistance and reducing the effectiveness of antibiotics in treating pathogenic infections.

The extent of antibiotic uptake by plants varies depending on the type of antibiotics and plants, and it is directly proportional to the amount of antibiotics used. Antibiotic residues have various adverse effects on plants, including impaired growth, fewer leaves, and lower chlorophyll content [150]. Cucumber, tomato, and lettuce treated with tetracycline and sulfonamides showed lower shoot and root weight compared to the control group [150].

Even at low doses, antibiotics can have a significant impact on plant characteristics. These effects—which include delayed germination, lower biomass, and post-germinative development—can reduce the output of farmland fertilized with manure treated with antibiotics. Herbs were found to be more responsive to antibiotics than grasses, with the effects depending on species and functional category [160]. Species-specific reactions to antibiotics could alter the species’ composition of natural communities on field margins, potentially affecting their ability to compete. Such species-specific reactions could also change the composition of the plant species’ community, indirectly affecting higher trophic level species, such as pollinating and herbivorous insects [160]. The toxicity of commercially available single antibiotics and antibiotics in combination was assessed with the root development of *Sinapis alba* L [161]. Sulfadiazine was found to be the most toxic, while tetracycline and enrofloxacin were found to be the least toxic [161].

The impact of antibiotics on higher plants can be both morphological and physiological, leading to breakdowns in chlorophyll production and damage to photosystems [160,162]. A study investigating the effect of nine antibiotics on *Triticum aestivum* found that penicillins, cephalosporins, and tetracyclines had an impact on the photosynthetic electron transport rate, while tetracyclines, ciprofloxacin, and erythromycin greatly reduced the content of photosynthetic pigments, including chlorophylls and carotenoids [163].

Veterinary antibiotics can also affect plant performance by being released into farming fields through grazing livestock or manure and being absorbed, stored, altered, or sequestered by plant metabolic processes. A study showed that penicillin, sulfadiazine, and tetracycline antibiotics can have an impact on the elemental components of plants, including macro- and micro-elements, with the most significant influence being produced by penicillin. Roots were found to be the most responsive to antibiotics compared to stems and leaves, and even at low concentrations, antibiotics in the soil could disrupt the scaling relationships between roots and other plant organs, which could affect the plant’s metabolic processes and overall performance [164].

#### 3.2.1. Impact on Public Health

AR is a major global health concern, as highlighted by several studies [111,112,113]. The widespread use of antibiotics necessitates research into their impact on microbiota and health. The gut microbiota plays a crucial role in maintaining not only intestinal but also overall health and can be disturbed by a variety of factors, including antibiotics. Antibiotic use can result in decreased microbial community, modifications to the functional characteristics of the microbiota, and the emergence and adaptation of antibiotic-resistant organisms, which can all negatively impact the health of hosts and make them more vulnerable to infection by pathogens [165].

Infant’s resistome profile and gut bacteria colonization can be affected by perinatal and peripartum antibiotic treatment, as evidenced by research [166]. Dams exposed to cefoperazone during pregnancy were found to have offspring with changed gut microbial populations and increased susceptibility to naturally occurring and chemically induced colitis [167]. Maternal antibiotic intake during pregnancy has also been linked to a change in the microbial makeup of infants [168,169]. Antibiotic use during pregnancy has been associated with functional impairment in development and cognition, obesity, immunological changes, and the onset of diabetes in children. Additionally, it has been linked to an increased risk of asthma and allergy in fetuses [170,171,172,173].

In adults, the use of antibiotics has been shown to cause changes in their gut and oral flora. For instance, a study found that ciprofloxacin treatment for 10 days reduced the abundance of *Bifidobacterium* but had no effect on the levels of *Lactobacillus* and *Bacteroides*, whereas clindamycin treatment for the same number of days caused *Lactobacillus* and *Bifidobacterium* to decrease, and *Bifidobacterium* did not normalize until one year after the end of treatment [174]. Figure 4 shows the effects of different antibiotics on the abundance of gut microbiota in humans.

Studies on healthy infants and children who have never been exposed to antibiotics have shown the existence of genes conferring resistance to β-lactams, fluoroquinolones, tetracycline, macrolides, sulfonamides, or numerous drug classes. *Enterococcus* spp., *Staphylococcus* spp., *Klebsiella* spp., *Streptococcus* spp., and *Escherichia/Shigella* spp. were found to be the main carriers of ARGs [175,176,177]. Bacteria in the gut can transmit genes both horizontally and vertically to similar and dissimilar bacteria because of their proximity and the ability of mobile genetic elements (MGEs) [178]. Because ARGs make treating infections more challenging, expensive, and ineffective, their presence in humans is a global concern. Moreover, since AR bacteria can be passed from mother to child through breastfeeding, ARGs in the gut microbiota of infants can come from those of their mothers [178].

Gut bacteria are responsible for the production of numerous essential metabolites, such as short-chain fatty acids (SCFAs) and amino acids, which have been identified in studies [179]. Antibiotics can affect the transcription of important functional genes, which code for the transport proteins, enzymes involved in carbohydrate metabolism, and proteins involved in protein synthesis [180,181]. This modification of gene transcription can result in bacterial resistance to various antibiotics and host defenses, leading to difficulties in treating human infections and equipment/pipe blockages in healthcare facilities and the food industry [129]. The functional changes induced by antibiotics in the microbiota lead to alterations in the microbial community and—consequently—the metabolites generated by the bacteria [182].

Bacteria use pattern recognition receptors (PRRs) to communicate with their hosts by producing signaling molecules, such as bile acids, SCFAs, fatty acids, lipopolysaccharide, lipoteichoic acid, flagellin, 5′—C—phosphate—G—3′ DNA, and peptidoglycan. These signaling molecules can interact with free fatty acid receptors, G-protein-coupled receptors, and nuclear receptors to provide energy to other cells and regulate immune cell function [183,184]. However, when bacteria are treated with antibiotics, PRRs such as Toll-like receptors are reduced, leading to downstream regulation of innate defenses [185].

Several studies have shown that antibiotics can cause direct harmful effects on host tissues, including oxidative tissue damage, mitochondrial damage, and reduced ribosomal gene expression [186,187]. The use of antibiotics has also been linked to an increased risk of breast cancer and miscarriage [188,189,190]. Antibiotics can directly affect host metabolism without the use of micro-organisms as a mediator. For example, high levels of AMP—which reduce the effectiveness of antibiotics and promote phagocytic activity—are among the host metabolite alterations, which are primarily local to the infection site. Antibiotics can also weaken immunological response due to the inhibition of immune cells’ respiratory activity [191].

#### 3.2.2. Impact on Environment

Various antibiotic classes are present in the environment, and the existence of antibiotic residues is determined by the pharmacokinetic profile of antibiotics [192]. The biological activity of antibiotics in different environmental matrices is determined by their bioavailability and interaction with environmental factors, such as pH, organic carbon content in soil, water type, and the type of organism present [192]. Understanding how antibiotics degrade in the environment is crucial. Antibiotics with a lower adsorption potential readily transfer into the aquatic environment, while those with a higher adsorption potential tend to accumulate and remain in soil [193,194]. Penicillins and cephalosporins tend to accumulate in sewage sludge and sediments, possibly forming complexes with cations, which could explain the bacterial cephalosporin resistance observed in sewage treatment facilities [163].

The effect of streptomycin and tetracycline on soil fertility and microbial activity varies depending on the soil type. Streptomycin reduces nitrification and soil fertility in humus-poor soil, whereas tetracycline reduces denitrification and jeopardizes soil microbial activity in humus-rich soil [195]. Both antibiotics increase microbial biomass while inhibiting the growth of white mustard seeds, indicating an increase in the allelopathic activity of micro-organisms in soil when antibiotics and their metabolites are present. Streptomycin has low solubility in humus-poor soils, posing a significant threat to agricultural productivity, particularly in low-fertility areas [195].

The spread of antibiotic-resistant genes (ARGs) in the environment is one of the most significant threats to human health and the environment. ARGs have been discovered in soil, freshwater and saltwater oceans, river water, the food chain, and even in humans (Table 5) [129,141,157,159,179,180]. Contamination of these environments with antibiotics and AR bacteria is driving the spread of bacterial resistance [196]. ARGs have also been found in viruses, in addition to bacteria [182]. Although some ARGs have historically been found in pristine or uncontaminated Antarctic soil habitats, widespread human usage of antibiotics is a significant contributor to their spread [183].

## 4. Biocontrol Agents to Control Plant Pathogens Rather Than Antibiotics

### 4.1. Use of Endophytes

#### 4.1.1. Use of Bacterial Endophytes

Bacterial endophytes have shown promising antagonistic activity against various food-borne and plant pathogenic bacteria. They achieve this by generating biocidal compounds, triggering the plant’s defense systems, or directly parasitizing the pathogen, thus limiting the proliferation of phytopathogens [229]. The most effective bacterial genera known for their broad-spectrum activity against micro-organisms are *Bacillus* and *Pseudomonas*, which produce different antimicrobial compounds effective against diverse fungal and bacterial plant pathogens [13,18]. In field tests, the application of both *Pseudomonas* and *Bacillus* strains as seed treatment improved tomato seed quality and significantly reduced the incidence of *C. michiganensis* bacterial canker [230,231].

In addition to *C. michiganensis*, other bacteria, such as *Streptomyces* sp. strain HL-12, *Bacillus subtilis*, *Trichoderma harzianum*, and *Rhodosporidium diobovatum*, have demonstrated antibacterial activity against *Pseudomonads* [232,233]. Notably, *Staphylococcus pasteuri* and *Staphylococcus warneri* showed promise as biocontrol agents against the bacterium *Xanthomonas citri* subsp. *citri*, which causes citrus bacterial canker [234].

Different strains of *Bacillus velezensis* have also shown efficacy against various fungal and bacterial pathogens. For example, *Bacillus velezensis* FZB42 has demonstrated high effectiveness against *Xanthomonas campestris* pv. *campestris* isolated from cabbage [235], while *Bacillus velezensis* IP22 isolated from fresh cheese was found to be highly active against *Xanthomonas euvesicatoria*, which causes pepper bacterial spots [236]. The antimicrobial activity of *Bacillus velezensis* is attributed to different lipopeptides and polyketides [235]. Furthermore, co-culture platform studies have revealed the efficacy of *Bacillus safensis* ZK-1 against the kiwifruit canker pathogen *P. syringae* pv. *actinidiae*, *Pseudomonas alcaligenes* ZK-2 against the turfgrass disease dollar spot pathogen *Clarireedia paspali*, and *Bacillus velezensis* ZK-3 against the rice bacterial blight pathogen *X. oryzae* pv. *oryzae* and rice blast fungus *Magnaporthe oryzae* [237].

Two strains of *Pantoea agglomerans*, PHYTPO1 and PHYTPO2, were found to be excellent biocontrol agents against tomato bacterial wilt caused by *R. solanacearum* (Smith) [35]. Additionally, the rhizosphere competence, effective biological control of tomato wilt symptoms in greenhouses, and effects on the native rhizosphere prokaryotic communities were examined for two bacterial strains, *Bacillus velezensis* (B63) and *Pseudomonas fluorescens* (P142), both of which displayed in vitro antagonistic activity toward *R. solanacearum* (B3B). Under field conditions, *B. velezensis* (B63) and *P. fluorescens* (P142) treatment significantly reduced wilt disease symptoms [238].

The ability of *Bacillus subtilis*, *Bacillus pumilus*, *Bacillus megaterium*, and *Pseudomonas fluorescens* to suppress *P. carotovorum* subsp. *carotovorum* was evaluated under in vitro and in vivo testing. *B. megaterium* was found to be quite efficient when used simultaneously or two hours after pathogen injection. Additionally, under artificially infected conditions, *B. pumilus* provided strong protection for the potato tuber, which was being preserved [239].

#### 4.1.2. Use of Fungal Endophytes

Endophytes can produce a range of antibacterial substances in addition to antifungal substances, which can protect the host plant against bacterial infections. These antibacterial substances can vary in their spectrum, with some offering broad-spectrum protection and others providing defense against specific types of bacteria [16]. Figure 5 shows the mechanisms of these biocontrol agents in killing PPB, plant growth promotion and stress response in addition to the antibacterial activity. Endophytic fungi can produce various secondary metabolites, including terpenoids, alkaloids, phenylpropanoids, aliphatic compounds, polyketides, acetol, hexanoic acid, acetic acid, and peptides, which have broad antibacterial properties [16,17]. Certain endophytes of the *Microdiplodia* genus, for example, can generate phomadecalin E and 8-acetoxyphomadecalin C—two terpenoids, which are effective against antagonistic strains of *Pseudomonas aeruginosa* [16]. Some strains of *P. aeruginosa* can cause soft root rot in plants such as *Panex ginseng*, *Arabidopsis*, and *Ocimum basilicum*, as well as opportunistic human infections [240,241].

In addition to *Microdiplodia*, *Chaetomium globosum* is another fungal endophyte, which produces broad-spectrum antibacterial substances with activity against several pathogenic bacteria and anti-biofilm properties [242]. Another fungal endophyte, *Trichoderma harzianum*, which was isolated from *Rosmarinus officinalis*, has shown significant antimicrobial activity against *P. aeruginosa*, *Staphylococcus aureus*, *Klebsiella pneumoniae*, *B. subtilis*, and *E. coli* [243]. In vitro and in vivo evaluations of the ability of *Trichoderma harzianum*, *Trichoderma viride*, and *Trichoderma virens* to inhibit *P. carotovorum* subsp. *carotovorum*-induced bacterial soft rot have also been conducted. When administered simultaneously or two hours before pathogen inoculation, *T. viride* and *T. virens* significantly reduced the symptoms of soft rot in potato tuber slices, which had been inoculated [239].

Recent research has suggested that the endophyte may be used as a biocontrol agent against phytopathogenic bacteria [17]. For example, *Diaporthe phaseolorum*, *Aspergillus fumigatus*, and *A. versicolor*—endophytes from healthy tomato (*Solanum lycopersicum*) plants—have been found to produce antibacterial metabolites, such as acetol, hexanoic acid, and acetic acid, which exhibit effective biocontrol activities against the tomato bacterial spot (*Xanthomonas vesicatoria*) [17]. Additionally, extracts from Cupressaceae hosts containing extracellular metabolites of endophytic *Aspergillus* spp. have been found to exhibit variable degrees of antibacterial activity against *Bacillus* sp., *E. amylovora*, and *P. syringae*, although the metabolites were not identified [244]. It is not clear whether the endophytic fungus directly creates these antimicrobial compounds, or whether the host plant produces them in response to endophyte inoculation [16]. Further research is needed to better understand the secretion of these chemicals and related gene expression [16].

### 4.2. Use of Viral Vectors to Encounter Pathogens

Phage-based biological control is a promising option for a specific and cost-effective control of plant pathogens [245]. Research has shown that most bacteriophages with antibacterial activity against phytopathogens originate from the *Podoviridae* and *Myoviridae* families [14,15,246,247,248,249]. A recent study demonstrated that a cocktail of bacteriophages belonging to the *Podoviridae* and *Myoviridae* families was effective against *Pectobacterium* species [250].

In addition, researchers have isolated new lytic phages—vRsoP-WF2, vRsoP-WM2, and vRsoP-WR2—from river water. These phages belong to the *Podoviridae* family and the T7likevirus genus, and they were effective in reducing the incidence of bacterial wilt caused by *R. solanacearum* in irrigation water and host plants [33]. Another phage, phiPccP-1—a member of the *Unyawovirus* genus and the *Studiervirinae* subfamily of the *Autographivirinae* family—demonstrated lytic activity against *Pectobacterium odoriferum* Pco14 and two other *Pectobacterium* species in Pyeongchang, South Korea. Furthermore, the application of phiPccP-1 significantly reduced the development of soft rot disease in the mature leaves of harvested Kimchi cabbage up to 48 h after Pco14 inoculation [251].

The Siphoviridae family also shows promising results, with bacteriophage CMP1 and CN77 demonstrating lytic activity against *C michiganensis*. A peptidase-active endolysin from these viruses selectively lysed *C michiganensis* without affecting other bacteria [252,253].

*X. oryzae* pv. *oryzae* is a major pathogen, which causes illnesses in rice worldwide, and recent studies have identified two *Myoviridae*-family bacteriophages—Xoo-sp13 and Xoo-sp14—which infect this bacterium in China, highlighting their potential as targets for phage therapy due to their wide host range [254,255]. Additionally, the filamentous phage XaF13 (*Inoviridae* family) isolated from Mexico was found to infect the phytopathogenic bacterium *Xanthomonas vesicatoria* [256], and the bacteriophage Xccϕ1 was evaluated for its effectiveness against the phytopathogen *X. campestris* pv. *campestris*, revealing the possible advantages of bacteriophages in modifying biofilm structure, decreasing bacterial growth, and manipulating plant metabolism and defense mechanisms [257].

Despite their potential benefits, the use of bacteriophages as biopesticides in agriculture is limited due to their unique biological characteristics [258]. The challenging registration procedure and negative characteristics associated with bacteriophages as active molecules have resulted in a limited number of bacteriophage-based biopesticides on the market. However, Table 6 presents a shortlist of other bacteriophages, which have shown significant antibacterial activity against phytopathogenic bacteria, with most of them belonging to the Podoviridae and Myoviridae families.

Overall, these studies suggest that phage-based biological control could be a useful tool in the fight against plant pathogens, and further research is needed to develop more effective and specific phage-based treatments.

### 4.3. Use of Genetically Modified Organisms

Genetically modified bacteria have been created and distributed with plants in experimental field plots to make it easier to track them and evaluate their effects on the local microbiome. The survival and expression of advantageous features of these bacteria in field environments are highly dependent on local environmental conditions. Another alternative approach for genetically modified organisms is the incorporation of genes responsible for encoding different antimicrobial peptides in plants. However, some microbes capable of biosynthesizing these peptides have not been used successfully in field trials. Studies have shown that transcription factors play a crucial role in establishing the sensory transcription regulatory networks necessary for plant immunity in response to pathogen-induced cellular responses [264].

Recent research has shown that the recombinantly generated endolysin from the CMP1 bacteriophage is completely resistant to *C. michiganensis* in tomato plants [265]. *Serratia marcescens* strain B2 has been found to effectively manage cyclamen and rice fungal infections, although abiotic and biotic variables can negatively impact its activity. To overcome this limitation, a genetically altered rice-indigenous bacteria were created by inserting *S. marcescens*-isolated genes and placing them under the control of various promoter types. These genetically modified bacteria successfully inhibited *Pyricularia oryzae*-caused rice blast disease and were unaffected by abiotic or biotic influences [266].

Tomato plants (*Solanum lycopersicum*) expressing the Bs2 resistance gene from pepper have demonstrated increased resistance to bacterial spot disease caused by *Xanthomonas* species in replicated multi-year field studies conducted under commercial-type growth settings. VF 36-Bs2 plants displayed the lowest disease severity among all tomato varieties studied, with the Bs2 gene decreasing disease to exceptionally low levels in the highly susceptible VF 36 variety. In addition, transgenic lines produced approximately 2.5 times as much commercial fruit as their non-transformed parent lines [267]. Overexpression of plant ferredoxin-like protein (PFLP) in transgenic plants has been known to provide remarkable resistance to various bacterial infections. The PFLP gene was introduced into *Arabidopsis*, and these transgenic plants demonstrated resistance to soft rot bacterial pathogen *P. carotovorum* subsp. *carotovorum* [268]. The *Fusarium oxysporum* resistance gene I-3 has also been introduced into cultivated tomato (*Solanum lycopersicum*) from wild tomato species [269].

## 5. Conclusions

The importance of monitoring the antibiotic resistance (AR) patterns of plant pathogens, including fungi, cannot be overstated. However, there is a lack of data on this topic, making regular surveillance essential. It is recommended that regulatory authorities conduct regular AR surveillance programs to provide guidance, regulate antibiotic use, and monitor antibiotic resistance in the field. Non-pathogenic strains should also be examined, as they can contribute to the spread of antibiotic resistance. Alternative methods, such as biocontrol agents, show great potential for controlling phytopathogens and should be implemented in full field trials with appropriate farmer training. Additionally, genetic engineering can be utilized to produce endophytes, which are effective against phytopathogens, providing another promising avenue for disease control in agriculture.

## Figures and Tables

**Figure 1 plants-13-01135-f001:**
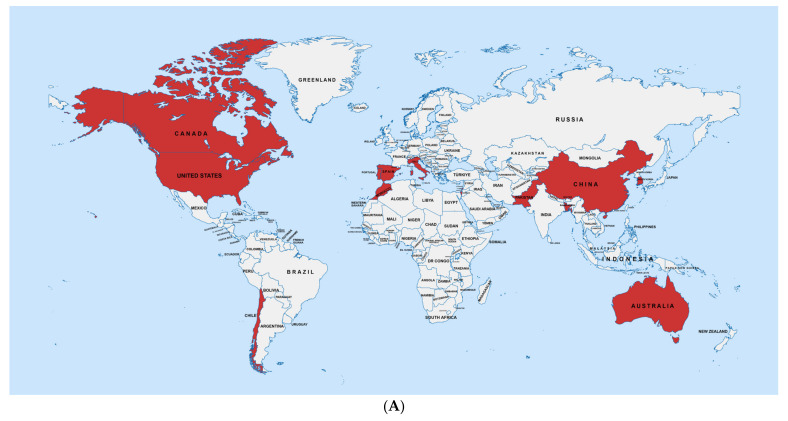
Locations where studies on the antibiotic resistance pattern of PPB were carried out (marked red) (**A**). Interconnected research landscape of the reviewed articles: Antibiotic resistance and bacterial pathogens in agriculture (**B**). The network was prepared using VOSviewer software, version 1.6.20.

**Figure 2 plants-13-01135-f002:**
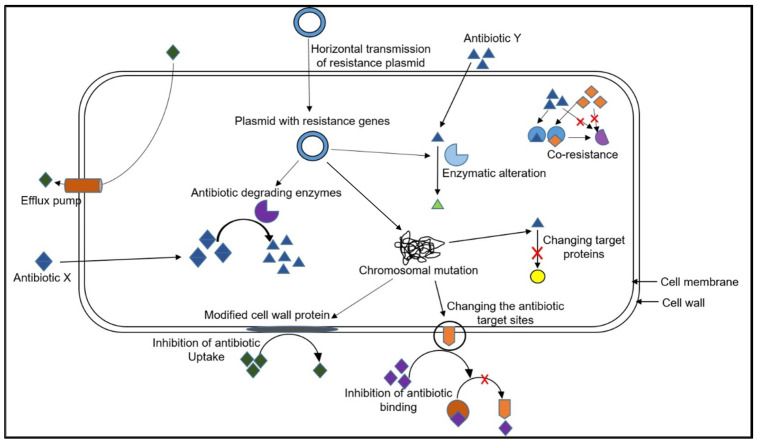
Overview of the different strategies bacteria use to resist antibiotics.

**Figure 3 plants-13-01135-f003:**
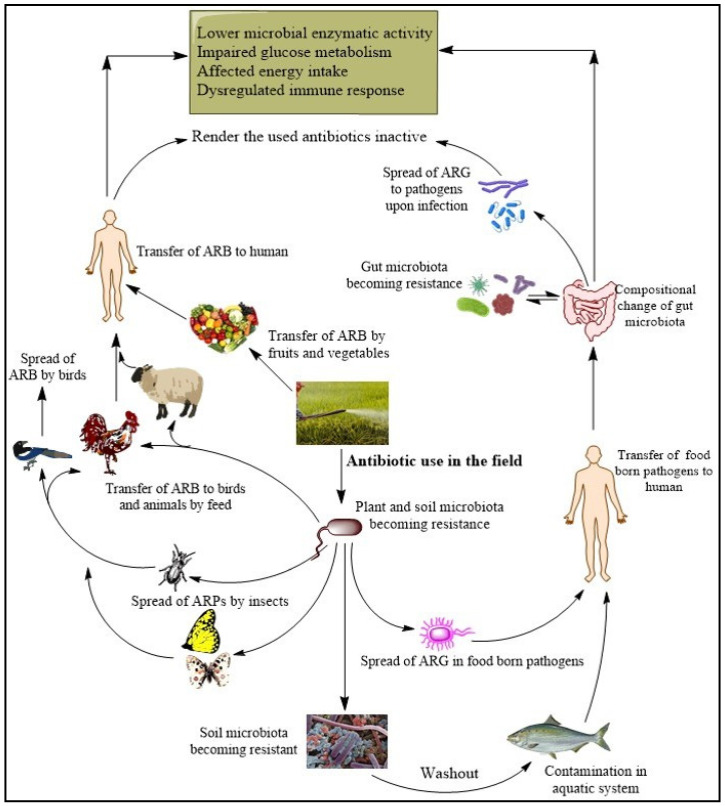
A diagram illustrating the widespread dissemination of antibiotic resistance throughout the food chain and its ultimate consequences for human health.

**Figure 4 plants-13-01135-f004:**
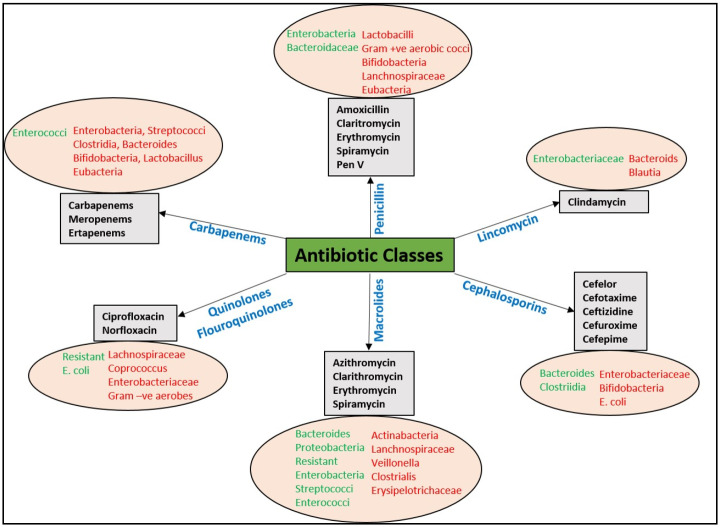
Effect of various antibiotics on the abundance of gut microbiota, with an increase indicated by the green color on the left, and a decrease indicated by the red color on the right (reproduced with permission from Wiley [165]).

**Figure 5 plants-13-01135-f005:**
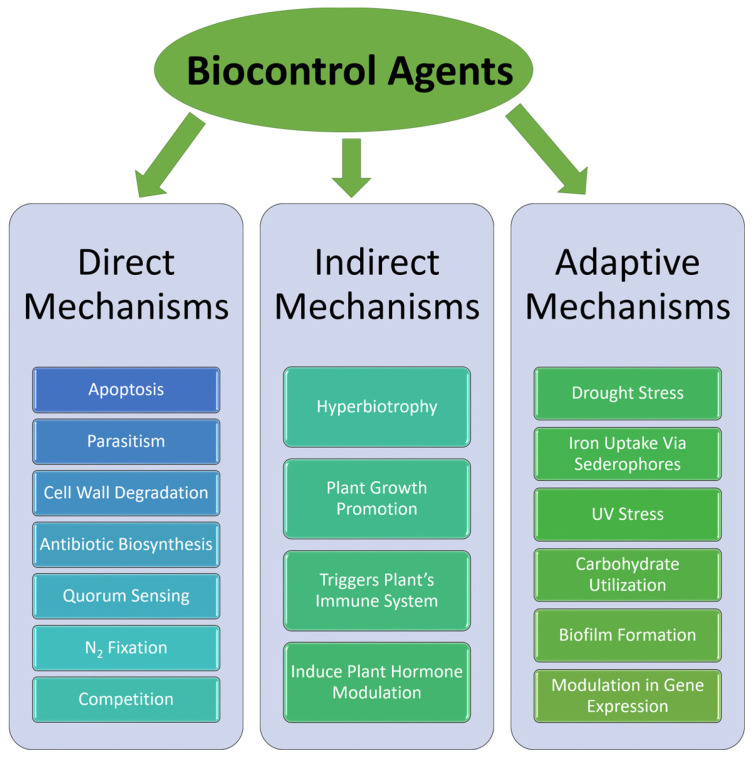
Summary of the mechanisms of biocontrol agents in promoting plant growth and providing protection against pathogens.

**Table 1 plants-13-01135-t001:** Crops in which antibiotics are mostly used to control bacterial diseases and their recommended antibiotics according to Verhaegen et al., 2023, and Verhaegen et al., 2024 [1,110].

Crop	Diseases	Causative Agent	Recommended Antibiotic
Rice	Bacterial panicle blight	*Burkholderia glumae*	Oxolinic acid, Streptocycline
	Bacterial leaf blight	*X. oryzae* pv. *oryzae*	
Tobacco	Wildfire	*P. syringae* pv. *tabaci*	Streptomycin
Tomato	Bacterial canker	*Clavibacter michiganensis* pv. *michiganensis*	Oxytetracycline, Gentamycin, Streptocycline
	Bacterial wilt	*R. solanacearum*	
	Bacterial speck	*P. syringae* pv. *tomato*	
Citrus	Citrus canker	*Xanthomonas axonopodis* pv. *citri*	Streptomycin
Paprika	Bacterial canker	*C. michiganensis* subsp. *capsici*	Streptomycin
Maize	Wilt and blight	*C. michiganensis* subsp. *nebraskensis*	Streptomycin
Potato	Blackleg	*P. atrosepticum*	Oxytetracycline, Gentamycin, Streptocycline
	Bacterial wilt	*R. solanacearum*	
	Soft rot	*Pectobacterium carotovorum*	
	Ring rot	*C. michiganensis* subsp. *sepedonicus*	
Eggplant	Bacterial wilt or southern wilt	*R. solanacearum*	Oxytetracycline, Gentamycin, Streptocycline
Cabbage	Bacterial black rot	*X. campestris* pv. *campestris*	Oxytetracycline, Gentamycin, Streptocycline
	Soft rot	*P. carotovorum*	
Watermelon	Black rot	*Xanthomonas* spp.	Gentamycin
Onion	Brown rot	*Pseudomonas aeruginosa*	Streptocycline

**Table 2 plants-13-01135-t002:** Antibiotic resistance pattern of PPB.

Antibiotic	Bacterial % of Resistance (Resistance/Total Samples)	Resistance Mechanism	Reference
Streptomycin	*P. syringae* 8.42% (8/95)	Probable chromosomal mutation	[31]
	*P. syringae* 5.26 (3/57)	Probable chromosomal mutation	[118]
	*X. smithii* subsp. *citri* 44.1%-88.7% (49/111-219/247)	Presence of the *str*B gene, chromosomal mutation	[117]
	*Xanthomonas oryzae* pv. *Oryzae* 26.67% (4/15)	Presence of the *aadA*1 gene	[119]
	*X. axonopodis* 55.5% (11/20)	Not mentioned	[120]
	*C. michiganensis* 1.68% (3/179)	Mutation in the *rps*L gene	[116]
	*C. michiganensis* subsp. *michiganensis* 84% (21/25)	Mutation in the *rps*L gene	[121]
	*Erwinia amylovora* 18.1% (20)	Not mentioned	[122]
	*X. arboricola* pv. *Pruni* 7.1% (7/99)	Presence of *str*AB genes	[123]
	*E. amylovora* 2.66% (34/1280)	Presence of the *str*A/*str*B gene	[124]
Tetracycline	*P. syringae* 1.01% (1/95)	Genetic modification	[31]
	*P. syringae* 3.5% (2/57)	Chromosomal mutation	[118]
	*Agrobacterium tumefaciens* 6.67% (02/30)	Not mentioned	[125]
	*A. tumefaciens* 100% (4/4)	Not mentioned	[126]
Oxytetracycline	*X. arboricola* pv. *Pruni* 7.1% (7/99)	Presence of *tet*C, *tet*R genes	[123]
Ampicillin	*P. syringae* 57.9% (55/95)	Not mentioned	[31]
	*P. syringae* 61.4% (35/57)	Chromosomal mutation	[118]
	*P. carotovorum* 22.73% (5/22)	Not mentioned	[127]
	*A. tumefaciens* 100% (30/30)	Not mentioned	[125]
Amoxicillin	*A. tumefaciens* 100% (30/30)	Not mentioned	[125]
Doxycycline	*A. tumefaciens* 13.34% (04/30)	Not mentioned	[125]
Copper	*P. syringae* 75% (69/92)	Not mentioned	[31]
Chloramphenicol	*P. syringae* 37.9% (36/95)	Not mentioned	[31]
	*P. syringae* 10.53% (6/57)	Chromosomal mutation	[118]
	*E. amylovora* 0% (20)	Not mentioned	[122]
Rifampicin	*P. syringae* 16.8% (16/95)	Not mentioned	[31]
	*A. tumefaciens* 100% (4/4)	Not mentioned	[126]
Kanamycin	*P. syringae* 1.01% (1/95)	Genetic modification	[31]
	*P. syringae* 0% (0/57)	Chromosomal mutation	[118]
	*A. tumefaciens* 0% (0/4)	Not mentioned	[126]
Gentamicin	*Ralstonia pickettii* 96.97% (32/33)	Presence of ICE- and ars-operon-related genes	[128]
	*X. axonopodis* 33.3% (20)	Not mentioned	[120]
	*E. amylovora* 9.99% (20)	Not mentioned	[122]
	*P. syringae* 1.75% (1/57)	Chromosomal mutation	[118]
Bacitracin	*X. axonopodis* 77.7% (20)	Not mentioned	[120]
	*E. amylovora* 81.89% (20)	Not mentioned	[122]
Cefotaxime	*X. axonopodis* 100% (20)	Not mentioned	[120]
	*E. amylovora* 100% (20)	Not mentioned	[122]
Cephalothin	*P. carotovorum* 22.73% (5/22)	Not mentioned	[127]
Cephradine	*A. tumefaciens* 26.66% (08/30)	Not mentioned	[125]
Cefuroxime	*A. tumefaciens* 10% (2/19)	Not mentioned	[129]
	*A. tumefaciens* 0% (0/4)	Not mentioned	[126]
Spectinomycin	*P. syringae* 3.5% (2/57)	Chromosomal mutation	[118]

**Table 3 plants-13-01135-t003:** Higher minimum inhibitory concentration (MIC) values of antibiotics found against PPB in previous studies.

Antibiotic	PPB	MIC50/MIC90 (µg/mL)	Lowest Observed MIC (µg/mL)	Highest Observed MIC (µg/mL)	Reference
Streptomycin	*X. oryzae* pv. *Oryzae*	≤100/300	1	300	[119]
	*C. michiganensis*	4/128	4	128	[116]
	*C. michiganensis* subsp. *michiganensis*	250/500	2	500	[9]
Ampicillin	*P. syringae*	6/6	6	6	[130]
Chloramphenicol	*P. syringae*	4/4	4	4	[130]
Colistin (polymyxin E)	*P. syringae*	0.094/0.094	0.094	0.094	[130]
Erythromycin	*P. syringae*	4/4	2	4	[130]
Kanamycin	*X. oryzae* pv. *Oryzae*	100/>100	1	>100	[119]
Netilmicin	*X. oryzae* pv. *Oryzae*	100/100	1	100	[119]
Sulfamethoxazole	*P. syringae*	192/192	192	192	[130]
Tetracycline	*P. syringae*	0.25/0.25	0.19	0.25	[130]
Gentamicin	*X. oryzae* pv. *Oryzae*	50/100	1	100	[119]
	*R. pickettii*	>256/>256	4	>256	[128]
Rifampicin	*X. oryzae* pv. *Oryzae*	5/10	0.1	10	[119]
Tobramycin	*X. oryzae* pv. *Oryzae*	10/50	1	50	[119]
Spectinomycin	*X. oryzae* pv. *Oryzae*	>500/>500	1	>500	[119]

Note: MIC50 represents the concentration of each antibiotic inhibiting 50% of the isolates; MIC90 represents the concentration of each antibiotic inhibiting 90% of the isolates.

**Table 4 plants-13-01135-t004:** Antibiotic compounds (ACs) and antibiotic-resistant genes (ARGs) detected in the edible parts of plants. All data pertaining to antibiotic-resistant genes in edible plants were taken from Marti et al., 2013 [144].

Sample Plant	Antibiotic Compounds	Amount Detected (µg/kg)	Antibiotic-Resistant Genes Detected	References on ACs Detection
Radish root	Gentamicin	0.051	*inc*P *ori*T, *inc*Q *ori*V, *int*3, *aad*(A), *str*(A), *str*(B), *sul*1, *erm*(B), *bla*_OXAIl_, *int*2, *tet*(A), *erm*(E), *bla*_CTX-M_, *bla*_VIM_, *bla*_TEM_, *erm*(F)	[145,146,147]
	Streptomycin	0.015		
	Oxytetracycline	8.3		
	Sulfadoxine	0.1–0.4		
	Lincomycin	0.9–3.1		
	Sulfamethazine	1.1		
Carrot	Sulfamethazine	<0.98	*inc*P, *ori*T, *inc*Q, *ori*V, *aad*(A), *str*(A), *str*(B), *sul*1, *erm*(C), *int*1, *tet*(A), *tet*(S), *sul*1, *erm*(B), *erm*(E), *bla*_VIM_, *bla*_TEM_, *qnr*(B), *tet*(B), *tet*(T), *bla*_OXA-20_	[145,148,149]
	Monensin	<3.44–4		
	Erythromycin	0–0.52		
	Chloramphenicol	0.96–3.99		
	Norfloxacin	2.52–6.54		
	Tetracycline	0–1.33		
	Sulfamethazine	0–0.37		
	Penicillins G & V	0.05–0.3		
Lettuce leaf	Tetracycline	1.35–1.85	*inc*P, *ori*T, i*nc*Q, *rep*B, *inc*W, *int*3, *tet*(A), *tet*(Q), *tet*(S), *aad*(A), *str*(A), *sul*1, *erm*(B), *bla*_OXA1_, *bla*_VIM_, *bla*_TEM_	[149,150,151,152]
	Chloramphenicol	0.86–2.72		
	Norfloxacin	2.88–7.43		
	Azithromycin	0.8–4		
	Ciprofloxacin	3.8–4		
	Kasugamycin	5–10		
	Streptomycin	5–10		
	Tetracycline	77–211		
	Oxytetracycline	35–318		
	Chlortetracycline	346–1364		
	Sulfamethazine	7813–25,993		
	Sulfamethoxazole	8582–30,589		
	Sulfadimethoxine	1773–7876		
Tomato	Tetracycline	199–1009	*tet*(T), *str*(A), *inc*P *ori*T, *inc*Y, *int*2, *int*3, *tet*(A), *tet*(S), *aad*(A), *str*(A), *str*(B), *erm*(B), *erm*(E), *bla*_CTX-M_, *bla*_VIM_, *bla*_TEM_, *tet*(T), *erm*(F), *bla*_PSE_, *bla*_OXA-20_	[148,150,151]
	Oxytetracycline	590–3231		
	Chlortetracycline	231–864		
	Sulfamethazine	9573–42,445		
	Sulfamethoxazole	17,193–38,467		
	Sulfadimethoxine	6113–20,887		
	Kasugamycin	5–10		
	Streptomycin	5–10		
	Penicillins G & V	0.05–0.3		
Cucumber	Tetracycline	89–496	*inc*P, *ori*T, *inc*P, *trf*A1, *str*(B), *sul*1, *erm*(B), *bla*_OXAII_	[148,150]
	Oxytetracycline	175–1603		
	Chlortetracycline	310–1320		
	Sulfamethazine	5359–16,319		
	Sulfamethoxazole	5633–11,330		
	Sulfadimethoxine	4924–12,692		
	Penicillins G & V	0.05–0.3		
Pepper	Chlortetracycline	<10	*int*3, *tet*(T), *str*(B), *sul*1, *vat*(B), *bla*_OXAII_	[145,153]
	Sulfamethazine	<10		
	virginiamycin	<10		
Mushroom	Tetracyclines	0.3–1.5		
	Sulfonamides	0.3–1.5		
	Penicyllins	0.3–3		
	Macrolides	0.3–3		
	Fluoroquinolones	0.3–1.5		
	cephalosporins	0.3–1.5		
Orange	Penicillin G	0.1–0.25		[148,154]
	Penicillins G & V	0.05–0.3		
Lemon	Penicillin G	0.1–0.25		[148,154,155]
	Penicillins G & V	0.05–0.3		
Grapefruit	Penicillin G	0.1–0.25		[154,155]

**Table 5 plants-13-01135-t005:** Antibiotic compounds (ACs) and antibiotic-resistant genes (ARGs) detected in different living forms and environments.

Samples	Antibiotic Compounds	Amount Detected (ng/kg)	Antibiotic-Resistant Genes Detected	References on ACs Detection	References on ARGs Detection
Drinking water or groundwater	Sulfapyridine	0.052	*tet*M, *tet*O, *tet*Q, *erm*F, *sul*1	[197,198,199,200]	[198,200]
	Sulfamethoxazole	0.3–18.6			
	Ciprofloxacin	0.4–224.4			
	Enrofloxacin	0.2–11.2			
	Norfloxacin	0.4–3.6			
	Florfenicol	3.3–26.1			
	Erythromycin	0.05			
River water or surface water	Sulfapyridine	0.2–3.1	*bla*_CTX_, *bla*_TEM_, *tet*A, *tet*B, *tet*M, *tet*W, *tet*O, *tet*Q, *tet*X, *erm*B, *erm*C, *erm*F, *aac(6′)-Ib-cr*, *qep*A, *qnr*S, *sul*1, *sul*2, *van*A, *mec*A, *amp*C	[196,197,198,199,200,201,202,203]	[198,200]
	Sulfamethoxazole	0.3–13.0			
	Ciprofloxacin	0.2–18.8			
	Enrofloxacin	0.2–52.2			
	Levofloxacin	0.3–6.0			
	Norfloxacin	0.2–78.1			
	Florfenicol	1.6–15.3			
	Doxycycline	1.9–3.5			
	Metronidazole	0.4–1.6			
	Erythromycin	0.1–1.7			
	Clarithromycin	0.35			
	Roxithromycin	1–913			
	Ofloxacin	2.2–2.9			
	Azithromycin	0.14–30.27			
	Norfloxacin	0.11–2200			
Wastewater	Sulfapyridine	0.4–2.2	*bla*_CTX_, *bla*_TEM_, *bla*_OXA_, *bla*_SHV_, *tet*A, *tet*B, *tet*M, *tet*W, *tet*O, *tet*Q, *tet*X, *erm*B, *erm*C, *erm*F, *aac(6′)-Ib-cr*, *qep*A, *qnr*S, *sul*1, *sul*2, *van*A, *mec*A, *amp*C	[197,198,199,200,201,203]	[198,200,204]
	Sulfamethoxazole	0.6–20.9			
	Ciprofloxacin	0.91–99.3			
	Enrofloxacin	0.86–3579.6			
	Levofloxacin	0.5–19,981.6			
	Norfloxacin	0.6–24.6			
	Chloramphenicol	0.99			
	Florfenicol	2.4–6.8			
	Doxycycline	1.8–264.4			
	Metronidazole	0.64–1.45			
	Ampicillin	900–1600			
	Erythromycin	up to 6.0			
	Clarithromycin	18–1800			
	Roxithromycin	32–1492			
	Ofloxacin	1210			
	Azithromycin	329.55			
	Norfloxacin	44.04–2900			
River sediment	Ciprofloxacin	0.16–21.74	*intI*1, *sul*2, *bla*_TEM_, *flo*R, *erm*B, *sul*1, *ere*A, *tet*W, *tet*M, *tet*C, *mec*A, *bla*_OXA-58_, *bla*_KPC-3_,	[94,97,98,99,100,101,104].	[205,206,207]
	Enrofloxacin	0.16–24.42			
	Levofloxacin	0.82–2.89			
	Norfloxacin	0.14–2.20			
	Chloramphenicol	0.98–1.53			
	Ciprofloxacin	0.68–112.7			
	Enrofloxacin	0.4–112.69			
	Levofloxacin	0.77–100.91			
	Norfloxacin	0.04–6600			
	Chloramphenicol	0.62–16.10			
	Doxycycline	1.44–57.32			
	Ampicillin	4600–43,800			
	Clarithromycin	330–9930			
	Azithromycin	43,000			
Soil	Ciprofloxacin	0.3–18.2	*aac(6′)-IB-CR*, *sul*2, *tet*A, *tet*X, *tet*W, *sul*1, *erm*F, *bla*_TEM_, *aad*A15, *aad*A13, *aad*A, *bla*_LCR-1_, *aac(3)-Ia*, *bla*_OXA-347_, *tet*C, *mef*C, *aph(6)-Ib*, *aad*A16, *dfr*A1, *aph(3′)-Ib*, *tet*M, *shv*, *erm*C,	[197]	[208,209,210]
	Enrofloxacin	0.4–5.5			
	Levofloxacin	0.2–6.5			
	Norfloxacin	0.2–4.6			
	Chloramphenicol	1.0–10.5			
	Doxycycline	1.1–5.5			
Human plasma, serum, urine, and feces	Ceftriaxone	1.01–200	*cfx*A, *aac*A, *erm*B, *erm*D, *tet*Q, *tet*W, *tet*O, *sul*2, *tet*32, *tol*C, *aad*A1, *bla*_SHV_, *bla*_SHV(156G)_, *acr*B, *bla*_SHV(238G240E)_, *tet*M, *omp*F, *erm*A, *mef*A, *tet*A, *tet*B, *bac*A, *van*R, *aad*E, *van*S, *tet*32, *mac*B, *bcr*A,	[211,212,213,214,215,216,217]	[204,218,219,220]
	Metronidazole	0.05–50			
	Amoxicillin	0.0015–0.015/50			
	Ampicillin	0.0015–0.015/50			
	Levornidazole	0.005–2.0			
	Linezolid	0.07/4.7 × 10^6^			
	Oxacillin	2–100 × 10^6^			
	Ceftazidime	2–100 × 10^6^			
	Piperacillin	2–100 × 10^6^			
Pig manure and fecal samples	Ciprofloxacin	0.33–13.71	*et*Q, *lnu*C, *tet*40, *aad*E, *erm*F, *tet*44, *erm*B, *cfx*A, *aph(3′)-IIIa*, *cfx*A6, *tet*X, *tet*L, *tet*W, *tet*O, *mef*A,	[197,202]	[120]
	Enrofloxacin	0.45–3.86			
	Levofloxacin	0.51–5.66			
	Norfloxacin	0.42–1057.60			
	Doxycycline	1.04–36.46			
	Metronidazole	0.33–0.42			
Poultry meat, manure, and fecal samples	Enrofloxacin	5.37–55.4	*aad*A, *aad*A2, *aad*A3, *str*B, *erm*B, *sul*2, *tet*K, *tet*M, *tet*W, *tet*X, *aac3-1*, *tet*A, *acr*A, *amp*C, *pKD13*, *tet*B, *erm*A, *aad*A1, *erm*C, *str*A, *mph*A, *aad*A14, *bla*_CARB8_, *bla*_CARB10_, *aph(3′)-Ic*, *aad*A24, *aph(6′)-ld*, *tet*39, *aph(3′)-IIa*, *lsa*E, *tet*L, *lnu*B, *van*A, *van*B, *van*C2	[204,221,222,223]	[113,224,225,226]
	Sulfadimethoxine	5.37–55.4			
	Sulfamerazine	5.37–55.4			
	Tylosin	5.37–55.4			
	Metronidazole	1.0			
	Chloramphenicol	0.4			
	Carbendazim	0.2			
	Diethofencarb	2.5			
	Sulfabenzamide	2.0			
	Erythromycin	1.0			
	Enrofloxacin	460			
	Doxycycline	50			
Fish farms	Chloramphenicol	0.0548 ± 0.0099	*mex*B, *mex*F, *mex*W, *mex*D, *acr*B, *opr*N, *ade*B, *mex*E, *emr*D, *mdt*G, *mdt*F, *tol*C, *bc*R, *acr*A, *mdt*H, *sul*1, *tet*32, *tet*M, *tet*O, *tet*T, *tet*W, *aad*A1, *aad*A2, *cat*A1, *emr*B, *mat*A, *mef*A, *msr*A	[204,227,228]	[135,136,137,138,139]
	Quinolones	1000–11,600			
	Sulfonamide	100–7000			
	Penicillin	11,000–20,400			
	Tetracyclines	300–1300			
	Ciprofloxacin	1000			

**Table 6 plants-13-01135-t006:** List of bacteriophages found to be effective against different phytopathogenic bacteria.

Bacterial Pathogens	Experimental Plant	Bacteriophage Used	Phage Family	Reference
*P. carotovorum* subsp. *carotovorum*, *P. wasabiae*	*Solanum tuberosum* (potato)	ΦPD10.3 and PD23.1	*Myoviridae*	[14]
*P. carotovorum* subsp. *carotovorum*	*Lactuca sativa* (lettuce)	PP1	*Podoviridae*	[249]
*P. atrosepticum*	*Solanum tuberosum* (potato)	Peat1 and phiM1	*Autographiviridae*	[259]
*Pectobacteriun*	*Brassica rapa* (Kimchi cabbage)	phiPccP-1	*Autographiviridae*	[251,260]
*Dickeya solani*	*Solanum tuberosum* (potato)	LIMEstone1 and LIMEstone2	*Myoviridae*	[247]
		ΦD1, ΦD10 and ΦD11		[248]
		ΦPD10.3 and PD23.1		[14]
*P. syringae* pv. *actinidiae*	*Actinidia deliciosa* (kiwifruit)	PPPL-1 and KHUφ38	*Podoviridae*	[261]
		KHUφ34	*Myoviridae*	[261]
		Φ6		[262]
*P. syringae* pv. *syringae*	*Prunus avium* (cherry)	Φ1215, Φ1226, 137, Φ358, Φ369)		[263]
*R. solanacearum*	*Musa acuminate* (banana)	M5 and M8	*Podoviridae*	[246]
	*Solanum lycopersicum* (tomato) and *Solanum tuberosum* (potato)	ɸsp1	*Myoviridae*	[15]

## Data Availability

The dataset is available from the corresponding author upon reasonable request.

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
