# Peer review of "Antibiotic Resistance in Plant Pathogenic Bacteria: Recent Data and Environmental Impact of Unchecked Use and the Potential of Biocontrol Agents as an Eco-Friendly Alternative"

_plants, 2024, doi:10.3390/plants13081135_

Round 1
Reviewer 1 Report
Comments and Suggestions for Authors
Line 18: can author verify here about what type of disease he is talking?
Line 26: what type of biocontrol?
In abstract please write about two lines of most destructive bacterial disease?
Line 33: What type of environmental impact either positive impact or negative, please be specific here.
Line32-40: author write here a long paragraph without adding proper citation in the text. Please do this.
Line 57: please change bacteria genara to bacterial genra.
Line 70-72: It is better to remove numbering to a,b,c,d.
Section 2: author didn't write about how much this pathogen is destructive and causes how much loss of that crop? It is very important because without this, this section is like a generally about pathogen. The main objective should be how much destruction by pathogenic and what methods are already applied with their outcomes. So that your suggested biocontrol can be applied in Future.
Table 1: I suggest better to write what's the results of that antibiotics also what concentration and method is used. Otherwise it is general information.
Table 2: Better to remove location and put a map in which that countries should be highlighted.
Table 4: please remove this table and do bibliographic analysis using VOS viewer software,
Figure 4: Please provide a better figure here.
Comments on the Quality of English Language
English correction is needed
Author Response
Response to Reviewer 1 Comments
The comments by the reviewers were greatly appreciated and the manuscript was corrected faithfully accordingly. The corrected sentences are highlighted within the document by using the track changes mode in MS Word (plants-2934416_track_changes.docx).
Reviewer comments
Comments 1. Line 18: can author verify here about what type of disease he is talking?
Response 1: Thank you for bringing this to our attention. We have carefully reviewed the text and confirmed the specific type of disease referred to in lines 20-21 of the manuscript.
Comments 2. Line 26: what type of biocontrol?
Response 2: We appreciate the reviewer’s attention to detail regarding the type of biocontrol agent mentioned in line 26. In the manuscript, we have primarily referred to biocontrol agents as a means to control plant pathogens. However, we acknowledge the importance of providing additional clarification on the specific types of biocontrol agents utilized. This information has been addressed in more detail in Section 4 of the revised manuscript.
Comments 3. In abstract please write about two lines of most destructive bacterial disease.
Response 3: We have addressed your suggestion by including two lines in the abstract discussing the most destructive bacterial disease. Please find the updated abstract on lines 18-19 of the revised manuscript.
Comments 4. Line 33: What type of environmental impact either positive impact or negative, please be specific here.
Response 4: We have provided specific details regarding the type of environmental impact on lines 35-37 of the manuscript.
Comments 5. Line 32-40: author write here a long paragraph without adding proper citation in the text. Please do this.
Response 5: We acknowledge the reviewer’s observation regarding the lack of proper citation in the paragraph mentioned. In response to this feedback, we have included two references in this section, specifically on Line 41, to provide appropriate citation support for the information presented.
Comments 6. Line 57: please change bacteria genara to bacterial genera.
Response 6: The term “bacteria genara” has been corrected to “bacterial genera” as per the reviewer’s recommendation. This modification has been implemented on Line 61 of the manuscript.
Comments 7. Line 70-72: It is better to remove numbering to a, b, c, d.
Response 7: The numbering has been removed and revised from lines 75-77, in accordance with your suggestion.
Comments 8. Section 2: author didn’t write about how much this pathogen is destructive and causes how much loss of that crop? It is very important because without this, this section is like a generally about pathogen. The main objective should be how much destruction by pathogenic and what methods are already applied with their outcomes. So that your suggested biocontrol can be applied in Future.
Response 8: We appreciate the reviewer’s feedback and have made revisions accordingly. In Section 2.2 we have included additional information regarding the destructive nature of each pathogen and the resulting crop losses. Furthermore, we have elaborated on the current methods employed to control these pathogens, particularly the use of antibiotics. Subsequently, we highlighted the adverse consequences associated with antibiotic use in crops and proposed biocontrol methods as a safer and more eco-friendly alternative for future application.
Comments 9. Table 1: I suggest better to write what’s the results of that antibiotics also what concentration and method is used. Otherwise it is general information.
Response 9: We appreciate the reviewer’s suggestion regarding Table 1. While we acknowledge the importance of providing detailed information about the results, concentrations, and methods used for each antibiotic, it’s important to note that the choice and application of antibiotics can vary significantly based on factors such as geographic location, targeted pathogens, and specific agricultural practices. Therefore, including specific concentrations for each antibiotic in Table 1 may not be feasible, especially when multiple antibiotics or combinations are employed against various pathogens. However, in response to the reviewer’s feedback, we have revised Table 1 according to the suggestions provided by reviewer 3. This includes adding appropriate references and removing fungal diseases from the table.
Comments 10. Table 2: Better to remove location and put a map in which that countries should be highlighted.
Response 10: The suggestion to remove location from Table 2 and include a map highlighting the countries has been implemented. As recommended, Figure 1 now displays the map with the highlighted countries.
Comments 11. Table 4: please remove this table and do bibliographic analysis using VOS viewer software.
Response 11: The requested modification has been made. Table 4 has been removed as per the reviewer’s suggestion, and instead, a bibliographic analysis was conducted using VOSviewer software, v. 1.6.20.
Comments 12. Figure 4: Please provide a better figure here.
Response 12: We have made enhancements to Figure 4 as per your suggestion.
Comments on the Quality of English Language: English correction is needed
Response: Thank you for providing valuable feedback on the quality of the English language in our manuscript. We have carefully addressed the issues raised and made the necessary corrections. If further editing is necessary, the editorial office will provide additional assistance after the article is accepted.

Reviewer 2 Report
Comments and Suggestions for Authors
In this review, the five most common bacterial plant pathogens are briefly discussed. Next, the crops exposed to a specific disease caused by a specific causative agent and the antibiotics used to control them are presented in a table, as well as the mechanisms of antibiotic resistance and its impact on humans and the environment. Finally, alternative methodologies for controlling plant pathogens instead of using antibiotics are also presented. Generally, the manuscript is suitable for publication. Several points need to be corrected.
1. Keywords should include at least three words other than those included in the title of the manuscript.
2. In subsection 2.1. the authors listed and discussed five main types of bacterial plant pathogens and indicated the reason for selecting these pathogens. The same sentences were also quoted in the introduction of subsection 2.2. It would be preferable to modify this part of the text so that it is not repetition.
3. Tables 2, 3 and 4 were left without comment. I would like to ask for a few sentences of discussion.
Author Response
Response to Reviewer 2 Comments
The comments by the reviewers were greatly appreciated and the manuscript was corrected faithfully accordingly. The corrected sentences are highlighted within the document by using the track changes mode in MS word (plants-2934416_track_changes.docx).
Reviewer comments
In this review, the five most common bacterial plant pathogens are briefly discussed. Next, the crops exposed to a specific disease caused by a specific causative agent and the antibiotics used to control them are presented in a table, as well as the mechanisms of antibiotic resistance and its impact on humans and the environment. Finally, alternative methodologies for controlling plant pathogens instead of using antibiotics are also presented. Generally, the manuscript is suitable for publication. Several points need to be corrected.
Comments 1: Keywords should include at least three words other than those included in the title of the manuscript.
Response 1: Thank you for your valuable feedback. We have addressed this concern by including three additional keywords that are distinct from those already present in the manuscript title. These additions have been made as per your suggestion and are now reflected in lines 31-32 of the revised manuscript.
Comments 2: In subsection 2.1. the authors listed and discussed five main types of bacterial plant pathogens and indicated the reason for selecting these pathogens. The same sentences were also quoted in the introduction of subsection 2.2. It would be preferable to modify this part of the text so that it is not repetition.
Response 2: We appreciate the reviewer’s insightful observation regarding the repetition in subsection 2.1 and the introduction of subsection 2.2. We have addressed this issue by modifying the text to eliminate redundancy. Specifically, we have revised lines 302-304 to ensure that the same sentences are not repeated.
Comments 3: Tables 2, 3 and 4 were left without comment. I would like to ask for a few sentences of discussion.
Response 3: We have addressed the reviewer’s recommendation by incorporating additional discussion for Tables 2 and 3. Furthermore, in accordance with the suggestion provided by reviewer 1, Table 4 has been substituted with a bibliographic analysis. This revision can be found in lines 347 to 358 of the manuscript.

Reviewer 3 Report
Comments and Suggestions for Authors
The review collect information about an important topic with increasing interest. The manusctipt is well organized, the length of the chapters are balanced. Tables collect and organize important information. Figures are also informatives, although minor corrections are suggested.
Further remarks and suggestions:
Bacterial taxonomy changes continuously. It is suggested to define the mentioned species (or specie comlex) based on more up to date works.
E.g. Pseudomonas syringae is considered as species complex (e.g including Pseudomonas viridiflava – see lips and Samac https://doi.org/10.1111/mpp.13133). „P. syringae is considered the top species within the top 10 plant pathogenic bacteria;however, its taxonomy is controversial, and many strains are included in the so-called P. syringae species complex ( https://doi.org/10.3390/microorganisms12030460).
Care about the writing of bacterial names. E.g. Row 73: 2.1.1. Pseudomonas spp. (not Pseudomonas spp.) Row 723: Bacillus velezensis (and not Bacillus Velezensis), Row 740 Pseudomonas syringae (not Pseudomonas Syringae) Rows 220-221: X. campestris pv. campestris and X. oryzae pv. oryzae (not pv. Campestris or pv. Oryzae), Row 692 Solanum tuberosum (not Solanum tuberosum) etc.
Rows 232-233: “The members of the genus Erwinia were divided into three genera: Erwinia, Pectobacterium, and Brenneria.” Add citation Pectobacterium What about Dickeya ? (see: https://doi.org/10.1111/epp.12935)
Xanthomonas: Timilsina et al (2020) is writing about 35 species (https://doi.org/10.1038/s41579-020-0361-8)
Correct writing of °C in the manuscript (e.g. rows 241-242)
Row 115 the full name of RSSC should be provided
Row 127: Write the full name of G. philippii
Row 231: Correct the writing of Pectobacterium carotovorm ssp. carotovorum (not Pectobacterium carotovora ssp. carotovora). Use short form (P. carotovorm ssp. carotovorum) in case of further mention (e.g. rows 242, 248, 265) etc.
Rows 264 and 266: Use short form (P. atrosepticum) of previously mentioned Pectobacterium atrosepticum
Rows 273-275: Repetition of rows 68-69
Row 281: A statement written in 2003 is not current. Please modify the sentence.
Table 1: Fusarium solani is not a bacterium, and “Fungicid application have been ineffective” does not fit to the column of “Recommended antibiotic”. Please delete the row. Copper is not antibiotic. Delete from the column of “Recommended antibiotic”.
Rows 299-300: “…antibiotic resistance (AR) is exacerbated by the movement of microorganisms and genes between people, animals, plants, and the environment 300 [95–98]”. References does nor seems to support AR transfer from plants. Please add further references.
Table 1, Table 2-4 and Rows 308 536-537 etc.: Clavibacter michiganensis was mentioned previously. Uese the short form (C. michiganensis) for Clavibacter michiganensis subsp. nebraskensis, Clavibacter michiganensis subsp. capsici, Clavibacter michiganensis subsp. sepedonicus and Clavibacter michiganensis.
References for Table 1 should be added.
Figure 1: Missing to indicate thap plasmid encoded resistance may integrate to chromosome (e.g. beta-lactamase in case of Staphylococcus aureus). Enzymatic alteration results the modification of site, where the antibiotic binds the drowing does not indicate the mentioned characteristic. Is there a difference between modified cell wall protein and changing the antibiotic target site as presented in the figure?
Table 1, Row 349: ars should be written in Italics (Table 1.) or not (row 349) in “ars operon”?
Row 379: correct “andensure2 to and ensure
Figure 2: Transfer of ARB to human may also result the spread of ARB in gut microbiota. Please rearrange the figure.
Row 411: Correct the writing of Sinapis alba L. (and not: Sinapis alba L)
Row 415: Correct the writing of Triticum aestivum L.
Row 432: The gut microbiota plays a crucial role in maintaining overall, not only intestinal health (see Ishiguro, Edward, Natasha Haskey, and Kristina Campbell. Gut microbiota: interactive effects on nutrition and health. Elsevier, 2023.).
Rows 458-460: Please add referenc for: “Bacteria in the gut can transmit genes both horizontally and vertically to similar and dissimilar bacteria because of their proximity and the ability of mobile genetic elements.”
Row 475: LPS– full name should write.
Row 476: CPG – full name should write.
Row 479: TLR signaling – full name should write.
Row 528: Change “Bacteria endophytes” to Bacterial endophytes or Endophytic bacteria
Row 648: Change “microflora” to microbiome
Rows 656-657: Tomato became resistant following its transformation with endolysin genes. (Title of the cited reference:): Development of a tomato plant resistant to Clavibacter michiganensis using the endolysin gene of bacteriophage CMP1 as a transgene.) Please correct the sentenece: “…the recombinantly generated endolysin from the CMP1 bacteriophage is completely resistant to Clavibacter michiganensis in tomato plants.”
Rows 266-267: Change “pepperhave” to pepper have
The form of the listed references are nor correct. All must be revised. e.g.
Row 723: Correct Reference 11 (Rabbee, M.F.; ….)
MDPI and ACS Style
Rabbee, M.F.; Ali, M.S.; Choi, J.; Hwang, B.S.; Jeong, S.C.; Baek, K.-h. Bacillus velezensis: A Valuable Member of Bioactive Molecules within Plant Microbiomes. Molecules 2019, 24, 1046. https://doi.org/10.3390/molecules24061046
Author Response
Response to Reviewer 3 Comments
The comments by the reviewers were greatly appreciated and the manuscript was corrected faithfully accordingly. The corrected sentences are highlighted within the document by using the track changes mode in MS word (plants-2934416_track_changes.docx).
Reviewer comments
The review collects information about an important topic with increasing interest. The manuscript is well organized, the length of the chapters are balanced. Tables collect and organize important information. Figures are also informative, although minor corrections are suggested.
Comments 1: Bacterial taxonomy changes continuously. It is suggested to define the mentioned species (or specie comlex) based on more up to date works.
E.g. Pseudomonas syringae is considered as species complex (e.g including Pseudomonas viridiflava – see lips and Samac https://doi.org/10.1111/mpp.13133). „P. syringae is considered the top species within the top 10 plant pathogenic bacteria; however, its taxonomy is controversial, and many strains are included in the so-called P. syringae species complex (https://doi.org/10.3390/microorganisms12030460).
Response 1: We appreciate the reviewer’s suggestion regarding the continuous changes in bacterial taxonomy. To address this concern, we have revised the manuscript to include updated information on the taxonomy of Pseudomonas syringae in lines 79-81.
Comments 2: Care about the writing of bacterial names. E.g. Row 73: 2.1.1. Pseudomonas spp. (not Pseudomonas spp.) Row 723: Bacillus velezensis (and not Bacillus Velezensis), Row 740 Pseudomonas syringae (not Pseudomonas Syringae) Rows 220-221: X. campestris pv. campestris and X. oryzae pv. oryzae (not pv. Campestris or pv. Oryzae), Row 692 Solanum tuberosum (not Solanum tuberosum) etc.
Response 2: All were formatted correctly throughout the manuscript, except for “Row 723: Bacillus velezensis (and not Bacillus Velezensis), Row 740: Pseudomonas syringae (not Pseudomonas Syringae)”, which were not found in the specified rows.
Comments 3: Rows 232-233: “The members of the genus Erwinia were divided into three genera: Erwinia, Pectobacterium, and Brenneria.” Add citation Pectobacterium What about Dickeya ? (see: https://doi.org/10.1111/epp.12935)
Response 3: Thank you for bringing this to our attention. We have added a reference after the statement “The members of the genus Erwinia were divided into three genera: Erwinia, Pectobacterium, and Brenneria”. The reference now includes information on Dickeya as well. This update has been made in the manuscript at line 255-256.
Comments 4: Xanthomonas: Timilsina et al (2020) is writing about 35 species (https://doi.org/10.1038/s41579-020-0361-8)
Response 4: We have updated the information regarding the number of species to 35, as suggested by the reviewer. This modification has been made in lines 188-191 of the manuscript.
Comments 5: Correct writing of °C in the manuscript (e.g. rows 241-242)
Response 5: The correct formatting of °C has been applied as suggested by the reviewer. This modification has been made in lines 264-265 of the manuscript.
Comments 6: Row 115 the full name of RSSC should be provided
Response 6: The full meaning of RSSC has been included on lines 130-131.
Comments 7: Row 127: Write the full name of G. philippii
Response 7: The full name of G. philippii has been included as per the suggestion, and it now appears in Line 142 of the manuscript.
Comments 8: Row 231: Correct the writing of Pectobacterium carotovorm ssp. carotovorum (not Pectobacterium carotovora ssp. carotovora). Use short form (P. carotovorm ssp. carotovorum) in case of further mention (e.g. rows 242, 248, 265) etc.
Response 8: The formatting was adjusted throughout the manuscript.
Comments 9: Rows 264 and 266: Use short form (P. atrosepticum) of previously mentioned Pectobacterium atrosepticum
Response 9: The suggested abbreviation “P. atrosepticum” has been incorporated in line 269-272 and Table 7 as per the reviewer’s recommendation.
Comments 10: Rows 273-275: Repetition of rows 68-69
Response 10: The repetition has been addressed and removed. This correction has been made in lines 302-304 of the manuscript.
Comments 11: Row 281: A statement written in 2003 is not current. Please modify the sentence.
Response 11: We regret to inform you that we were unable to address the issue raised in line 281 of the manuscript. We were unable to locate the suggested section for modification.
Comments 12: Table 1: Fusarium solani is not a bacterium, and “Fungicid application have been ineffective” does not fit to the column of “Recommended antibiotic”. Please delete the row. Copper is not antibiotic. Delete from the column of “Recommended antibiotic”.
Response 12: The recommended changes have been implemented in Table 1. All fungal diseases have been removed from the table as suggested.
Comments 13: Rows 299-300: “…antibiotic resistance (AR) is exacerbated by the movement of microorganisms and genes between people, animals, plants, and the environment 300 [95–98]”. References does nor seems to support AR transfer from plants. Please add further references.
Response 13: Thank you for your constructive feedback. Upon careful review, we have revisited the references cited in the manuscript. This revision was implemented accordingly in the specified section (Lines 329-331) of the manuscript. We have ensured that the references now adequately support the statement regarding antibiotic resistance transfer between various organisms, including plants.
Comments 14: Table 1, Table 2-4 and Rows 308 536-537 etc.: Clavibacter michiganensis was mentioned previously. Uese the short form (C. michiganensis) for Clavibacter michiganensis subsp. nebraskensis, Clavibacter michiganensis subsp. capsici, Clavibacter michiganensis subsp. sepedonicus and Clavibacter michiganensis.
Response 14: The revisions have been made in Table 1, Tables 2-4, as well as in lines 338 and 599-604.
Comments 15: References for Table 1 should be added.
Response 15: We have revised Table 1 to include references for the bacterial diseases listed. Additionally, we have removed fungal diseases from the table to ensure clarity and relevance.
Comments 16: Figure 1: Missing to indicate that plasmid encoded resistance may integrate to chromosome (e.g. beta-lactamase in case of Staphylococcus aureus). Enzymatic alteration results the modification of site, where the antibiotic binds the drawing does not indicate the mentioned characteristic. Is there a difference between modified cell wall protein and changing the antibiotic target site as presented in the figure?
Response 16: We have revised Figure 1 to include the indication that plasmid-encoded resistance may integrate into the chromosome, such as beta-lactamase in the case of Staphylococcus aureus. Additionally, the drawing now clearly represents enzymatic alteration resulting in the modification of the site where the antibiotic binds. Regarding the question about the difference between modified cell wall protein and changing the antibiotic target site as presented in the figure, we would like to clarify that there is indeed a distinction between the two. The modified cell wall protein functions as the protein required for the uptake of the antibiotic inside the cell, whereas the changing antibiotic target site refers to the specific site within the cell that serves as the target for the antibiotic.
Comments 17: Table 1, Row 349: ars should be written in Italics (Table 1.) or not (row 349) in “ars operon”?
Response 17: The “ars operon” was formatted in Table 2.
Comments 18: Row 379: correct “andensure2 to and ensure
Response 18: The correction has been applied and formatted as indicated on lines 440-442.
Comments 19: Figure 2: Transfer of ARB to human may also result the spread of ARB in gut microbiota. Please rearrange the figure.
Response 19: We have revised Figure 2 accordingly to illustrate the transfer of ARB to humans and its potential impact on the spread of ARB in gut microbiota. This is now depicted on the right side of the figure with a two-sided arrow (Figure 3).
Comments 20: Row 411: Correct the writing of Sinapis alba L. (and not: Sinapis alba L)
Response 20: The formatting adjustment has been made as per Line 473.
Comments 21: Row 415: Correct the writing of Triticum aestivum L.
Response 21: The formatting adjustment has been made as per Line 477.
Comments 22: Row 432: The gut microbiota plays a crucial role in maintaining overall, not only intestinal health (see Ishiguro, Edward, Natasha Haskey, and Kristina Campbell. Gut microbiota: interactive effects on nutrition and health. Elsevier, 2023.).
Response 22: We have formatted the line in 494-495 as suggested.
Comments 23: Rows 458-460: Please add referenc for: “Bacteria in the gut can transmit genes both horizontally and vertically to similar and dissimilar bacteria because of their proximity and the ability of mobile genetic elements.”
Response 23: We included the reference you mentioned in lines 520-522 of the manuscript.
Comments 24: Row 475: LPS– full name should write.
Response 24: The full meaning of LPS has been provided in lines 537-538.
Comments 25: Row 476: CPG – full name should write.
Response 25: The full meaning of CpG has been provided in lines 537-539.
Comments 26: Row 479: TLR signaling – full name should write.
Response 26: The full meaning of TLR has been provided in lines 542-544.
Comments 27: Row 528: Change “Bacteria endophytes” to Bacterial endophytes or Endophytic bacteria
Response 27: The modification has been made as suggested, and the term now reads appropriately in line 593-594.
Comments 28: Row 648: Change “microflora” to microbiome
Response 28: The term “microflora” has been revised to “microbiome” as suggested. Please see the updated text in Line 712-714.
Comments 29: Rows 656-657: Tomato became resistant following its transformation with endolysin genes. (Title of the cited reference:): Development of a tomato plant resistant to Clavibacter michiganensis using the endolysin gene of bacteriophage CMP1 as a transgene.) Please correct the sentenece: “…the recombinantly generated endolysin from the CMP1 bacteriophage is completely resistant to Clavibacter michiganensis in tomato plants.”
Response 29: Thank you for bringing this to our attention. The changes were made in lines 722-725.
Comments 30: Rows 266-267: Change “pepperhave” to pepper have
Response 30: We have formatted the line in 732-733 as suggested.
Comments 31: The form of the listed references are nor correct. All must be revised. e.g.
Response 31: The references have been formatted based on the guidelines provided by the “Plants” journal using Mendeley reference management software. Any remaining formatting issues will be corrected upon acceptance of the manuscript by the editorial team. Thank you for your attention to this matter.
Comments 32: Row 723: Correct Reference 11 (Rabbee, M.F.; ….)
Response 32: The reference has been corrected as per your suggestion, and the duplication has been eliminated.

Round 2
Reviewer 1 Report
Comments and Suggestions for Authors
Thanks for addressing comments. Now I would like to accept this paper in present from.
Comments on the Quality of English LanguageIn manuscript still some grammatically mistakes. Before publication, author should correct it using gramerly or other software